



Sources and characteristics of size-resolved particulate organic acids and methanesulfonate in a
coastal megacity: Manila, Philippines
Connor Stahl[1], Melliza Templonuevo Cruz[2,3], Paola Angela Bañaga[2,4], Grace Betito[2,4], Rachel A.
Braun[1], Mojtaba Azadi Aghdam[1], Maria Obiminda Cambaliza[2,4], Genevieve Rose Lorenzo[2,5],
Alexander B. MacDonald[1], Miguel Ricardo A. Hilario[2], Preciosa Corazon Pabroa[6], John Robin
Yee[6], James Bernard Simpas[2,4], Armin Sorooshian[1,5]
[1]Department of Chemical and Environmental Engineering, University of Arizona, Tucson,
Arizona, 85721, USA
[2]Manila Observatory, Quezon City, 1108, Philippines
[3]Institute of Environmental Science and Meteorology, University of the Philippines, Diliman,
Quezon City, 1101, Philippines
[4]Department of Physics, School of Science and Engineering, Ateneo de Manila University,
Quezon City, 1108, Philippines
[5]Department of Hydrology and Atmospheric Sciences, University of Arizona, Tucson, Arizona,
85721, USA
[6] Philippine Nuclear Research Institute - Department of Science and Technology,
Commonwealth Avenue, Diliman, Quezon City, 1101, Philippines
*Correspondence to: armin@email.arizona.edu*



**Abstract**
A 16-month (July 2018 – October 2019) dataset of size-resolved aerosol composition is used to
examine the sources and characteristics of five organic acids (oxalate, succinate, adipate,
maleate, phthalate) and methanesulfonate (MSA) in Metro Manila, Philippines. As one of the
most polluted megacities globally, Metro Manila offers a view of how diverse sources and
meteorology impact the relative amounts and size distributions of these species. A total of 66
sample sets were collected with a Micro-Orifice Uniform Deposit Impactor (MOUDI), of which
54 sets were analyzed for composition. Organic acids and MSA surprisingly were less abundant
than in other global regions that are also densely populated. The combined species accounted for
an average of $0.80 \pm 0.66$ % of total gravimetric mass between 0.056 and 18 µm, leaving still
33.74 % of mass unaccounted for after considering black carbon and water-soluble ions and
elements. The unresolved mass is suggested to consist of non-water-soluble metals as well as
both water-soluble and non-water-soluble organics. Oxalate was approximately an order of
magnitude more abundant than the other five species ($148.59 \pm 94.26$ ng m$^{-3}$ versus others being
< 10 ng m$^{-3}$). Both PMF and correlation analysis is conducted with tracer species to investigate
the possible sources for organic acids and MSA. Enhanced biomass burning influence in the
2018 southwest monsoon (SWM18) resulted in especially high levels of submicrometer
succinate, MSA, oxalate, and phthalate. Peculiarly, MSA had negligible contributions from
marine sources but instead was linked to burning and combustion. Enhanced precipitation during
the two SWM seasons (8 June – 4 October 2018 and 14 June – 7 October 2019) coincided with
stronger influence from local emissions rather than long-range transport, leading to notable
concentration enhancements in both the sub- and supermicrometer ranges for some species (e.g.,
maleate and phthalate). While secondary formation via gas-to-particle conversion largely
explained submicrometer peaks for all species, several species (i.e., phthalate, adipate, succinate,
oxalate) exhibited a prominent peak in the coarse mode, largely owing to their association with
crustal emissions (i.e., more alkaline aerosol type) rather than sea salt. Oxalate's strong
association with sulfate in the submicrometer mode supports an aqueous-phase formation
pathway for the study region, but also high concentration during periods of low rain and high
solar radiation indicates photo-oxidation is an important formation pathway.



## 1. Introduction

Organic acids are ubiquitous components of ambient particulate matter and can contribute appreciably to total mass concentrations in diverse regions ranging from the Arctic to deserts (e.g. Barbaro et al., 2017; Ding et al., 2013; Duarte et al., 2017; Gao et al., 2003; Kondo et al., 2011; Skyllakou et al., 2017; Sun et al., 2012; Youn et al., 2013). Furthermore, another class of species contributing to ambient aerosol mass is organosulfur compounds, with methanesulfonate (MSA) being an example species (Bardouki et al., 2003b; Ding et al., 2017; Falkovich et al., 2005; Kerminen et al., 1999; Maudlin et al., 2015; Ziemba et al., 2011). The spatiotemporal and size-resolved mass concentration profiles of organic and sulfonic acids are difficult to characterize and can significantly vary depending on the time of day, season, region, and meteorological profile (Adam et al., 2020; Bagtasa et al., 2019; Kobayashi et al., 2004; Maudlin et al., 2015; Mochida et al., 2003; Reid et al., 2013). It is necessary to quantify their relative abundances, and to understand factors affecting their production and eventual removal to be able to quantify their influence on aerosol hygroscopic and optical properties (Beaver et al., 2008; Cai et al., 2017; Freedman et al., 2009; Marsh et al., 2017; Marsh et al., 2019; Myhre and Nielsen, 2004; Peng et al., 2016; Xue et al., 2009). Low molecular weight organic acids are water-soluble and can range widely in hygroscopicity when in their pure salt form depending on factors such as carbon number (Prenni et al., 2001; Saxena and Hildemann, 1996; Sorooshian et al., 2008) and interactions with other components in multi-component aerosol particles (Drozd et al., 2014). Organic acids are generally believed to effectively scatter light and have a cooling effect on climate (McGinty et al., 2009; Myhre and Nielsen, 2004), although their overall impact on properties such as refractive index in multicomponent aerosols is poorly characterized. Refractive indices for species investigated in this work range widely from 1.43 (MSA) to 1.62 (phthalic acid). MSA is assumed to be purely scattering similar to sulfate (Hodshire et al., 2019) and to have hygroscopic properties close to those of ammonium sulfate (Asmi et al., 2010; Fossum et al., 2018). However, its hygroscopic and optical behavior is not fully understood, and is still an active area of research (Liu et al., 2011; Peng and Chan, 2001; Tang et al., 2019; Tang et al., 2015; Zeng et al., 2014).

Decades of research into atmospheric organic acids and MSA have yielded rich insights into their sources, production mechanisms, and fate in the atmosphere (Baboukas et al., 2000; Bardouki et al., 2003a; Gondwe et al., 2004; Kawamura and Bikkina, 2016; Limbeck et al., 2001; Norton et al., 1983; Ovadnevaite et al., 2014; Sorooshian et al., 2009; van Pinxteren et al., 2015). MSA is produced predominantly from the oxidation of dimethylsulfide (DMS) emitted from oceans (Bates et al., 2004; Davis et al., 1998; Kerminen et al., 2017), but it also can be linked to biomass burning, urban, and agricultural emissions (Sorooshian et al., 2015). Sources of organic acids include primary emissions from biomass burning, biogenic activity, and the combustion of fossil fuels (Kawamura and Kaplan, 1987) and secondary formation via gas-to-particle conversion processes stemming from both biogenic (Carlton et al., 2006) and anthropogenic emissions (Sorooshian et al., 2007b). Secondary processing can include both aqueous phase chemistry in clouds (Blando and Turpin, 2000; Ervens, 2018; Ervens et al., 2014; Hoffmann et al., 2019; Rose et al., 2018; Sareen et al., 2016; Warneck, 2005) and photo-oxidation of volatile organic compounds (VOCs) in cloud-free air (Andreae and Crutzen, 1997;



Gelencsér and Varga, 2005). These various sources and production pathways result in mono- and
dicarboxylic acids being prevalent across a range of aerosol sizes (Bardouki et al., 2003b;
Kavouras and Stephanou, 2002; Neusüss et al., 2000; Yao et al., 2002). Little is reported in terms
of the size-resolved nature of organic acids and MSA over long periods of time with high
sampling frequency. Although insights have already been gathered from size-resolved
measurement studies (Table S1), most measurement reports are based on bulk mass
concentration measurements (Chebbi and Carlier, 1996; Kawamura and Bikkina, 2016).
Studying the seasonal variations of size-resolved organic acid and MSA aerosols could prove
vital in improved understanding of their formation and removal mechanisms, and associated
sensitivity to seasonally dependent sources and meteorological factors.
The Philippines is an important region to study aerosols due to the wide range in both
meteorological conditions and diverse local and regional emissions sources (Alas et al., 2018;
Bagtasa and Yuan, 2020; Braun et al., 2020; Hilario et al., 2020a; Kecorius et al., 2017). In
addition to aerosol sources from nearby regions (Hilario et al., 2020b), the Philippines also has a
significant source of local pollution largely consisting of vehicular emissions due to high
population density (Madueño et al., 2019), the use of outdated vehicles (Biona et al., 2017), ship
exhaust from high density shipping lanes (Streets et al., 1997; Streets et al., 2000), and more
lenient air regulations leading to significant air pollution due to rapid growth and urbanization
(Alas et al., 2018; Kecorius et al., 2017). This leads to Metro Manila containing some of the
highest black carbon (BC) concentrations in Southeast Asia, and quite possibly the world (Alas
et al., 2018; Hopke et al., 2011; Kecorius et al., 2017; Kim Oanh et al., 2006). Past aerosol
characterization work for that region has focused mainly on gravimetric analysis for total bulk
mass (e.g., $PM_{2.5}$, $PM_{10}$) (Bagtasa et al., 2018; Bagtasa et al., 2019; Cohen et al., 2009; Kim
Oanh et al., 2006), water-soluble inorganic and organic ion speciation (AzadiAghdam et al.,
2019; Braun et al., 2020; Cruz et al., 2019; Kim Oanh et al., 2006; Simpas et al., 2014; Stahl et
al., 2020b), and BC analysis (Alas et al., 2018; Bautista et al., 2014; Kecorius et al., 2017;
Takahashi et al., 2014). In an analysis of two size-resolved aerosol sets in Manila, a significant
portion of the total mass unaccounted for by the water-soluble inorganic, water-soluble organic,
and BC components was attributed to (but not limited to) organics and non-water soluble metals
(Cruz et al., 2019). However, a concentrated effort to characterize the contributions of the water-
soluble organic acids to the total aerosol mass in Manila over the course of a full year has not
been undertaken.
The aim of this study is to use a 16 month-long dataset of size-resolved composition in Quezon
City in Metro Manila to address the following questions: (i) how much do organic acids and
MSA contribute to the region's aerosol mass concentrations?; (ii) what are the seasonal
differences in the mass size distribution profile of organic acids and MSA, and what drives the
changes?; and (iii) what are the sources and predominant formation mechanisms of these species
in the sub- and super-micrometer diameter ranges? The results of this study are put in broad
context by comparing findings to those in other regions.

**2. Methods**



2.1 Study site description
Metro Manila is comprised of 16 cities and a municipality totaling to a population of about 12.88
million people and a collective population density of 20,800 $km^{-2}$ (Alas et al., 2018; PSA, 2016).
Quezon City is the most populated city in Metro Manila containing 2.94 million people with a
population density of 18,000 $km^{-2}$ (PSA, 2016), which is amidst the highest in the world.
Because of these reasons, Metro Manila is a quintessential location for examining locally
produced anthropogenic aerosols superimposed on a variety of other marine and continentally
influenced air masses transported from upwind regions (Kim Oanh et al., 2006).
Measurements were conducted over a 16-month period between July 2018 and October 2019 at
Manila Observatory (MO; 14.64° N, 121.08° E) on the third floor (~85 m a.s.l.) of an office
building, which is on the Ateneo de Manila University campus in Quezon City, Philippines (Fig.
1). Sampling was conducted approximately 100 m away from the nearest road on campus and
therefore campus emissions do not impact sampling to a large degree, qualifying the monitoring
site as an urban mixed background site (Hilario et al., 2020a) capturing local, regional, and long-
range transported emissions. The following four seasons were the focus of the sampling period:
the 2018 southwest monsoon (SWM18, 8 June – 4 October 2018) (PAGASA, 2018a, c), a
transitional period (Trans, 5 – 25 October 2018), the northeast monsoon (NEM, 26 October 2018
– 13 June 2019) (PAGASA, 2018b), and the 2019 southwest monsoon (SWM19, 14 June – 7
October 2019) (PAGASA, 2019b, a). These seasons have also been defined in other works (i.e.,
Akasaka et al., 2007; Cruz et al., 2013; Matsumoto et al., 2020) and can predominately be
separated into two general seasons, wet (SWM) and dry (NEM). Generally, there is a second
transitional period in May that transitions between the NEM and SWM (Bagtasa and Yuan,
2020), however, recent studies suggest that the transition is abrupt (Matsumoto et al., 2020).
Consequently, the second transitional period was combined with the NEM season.
2.2 Instrument description
Ambient aerosol was collected with a Micro-Orifice Uniform Deposit Impactor II (MOUDI II
120R, MSP Corporation, Marple et al. (2014)) using Teflon substrates (PTFE membrane, 2 μm
pores, 46.2 mm diameter, Whatman). The MOUDI-II is a 10-stage impactor with aerodynamic
cutpoint diameters ($D_p$) of 10, 5.6, 3.2, 1.8, 1.0, 0.56, 0.32, 0.18, 0.10, and 0.056 μm with a
nominal flow rate of ~30 L $min^{-1}$. A total of 66 MOUDI sets were collected on a weekly basis
usually over a 48-hour period; however, only 54 sets where analyzed for ions and 47 of those
sets were also analyzed for elements. A 48-hour period was chosen because it offered an optimal
compromise between gathering samples with fine temporal resolution and samples with a
sufficiently large chemical signal to exceed analytical limits of detection. Details of the sample
sets are shown in Table S2 can be found in more detail in Stahl et al. (2020b).
Water-soluble organic acids, MSA, and inorganic ions were speciated and quantified using ion
chromatography (IC; Thermo Scientific Dionex ICS-2100 system) with a flowrate of 0.4 mL
$min^{-1}$. The anionic species of relevance to this study were MSA, chloride ($Cl^-$), nitrate ($NO_3^-$),
sulfate ($SO_4^{2-}$), adipate, succinate, maleate, oxalate, and phthalate. These anions were resolved
using potassium hydroxide (KOH) eluent, an AS11-HC 250 mm column, and an AERS 500e





suppressor. The cationic species of relevance to this study was sodium ($Na^+$), which was detected
using methanesulfonic acid eluent, a CS12A 250 mm column, and a CERS 500e suppressor. The
IC instrument methods for anion and cation analysis can be found in Stahl et al. (2020b). Water-
soluble elements were measured using a triple quadrupole inductively coupled plasma mass
spectrometry (ICP-QQQ; Agilent 8800 Series). The quantified elements of relevance to this
study include Al, As, Cd, K, Ni, Pb, Rb, Ti, and V. Limits of detection (LOD) and recoveries
were calculated for all ionic and elemental species and provided in Table S3. Aside from the
species that are the focus of this study (organic acids and MSA), the other elements and ions
were included as they are useful tracers for different aerosol sources to aid in source
apportionment. Although pyruvate was speciated with IC, it is not considered with the other
organic acids because it was below the LOD for 48 of the 54 sets. It should also be noted that
only a subset of species were listed here, which were used for analyses. The full suite of species
can be seen in Stahl et al. (2020b).
Eleven of the 66 MOUDI sets included simultaneously operated MOUDIs next to each other to
complement the chemical speciation analysis with gravimetric analysis. A Sartorius ME5-F
microbalance (sensitivity of ± 1 µg) was used in an air-buffered room with controlled
temperature (20 – 23 °C) and relative humidity (RH: 30 – 40 %). Each substrate was passed near
an antistatic tip for approximately 30 seconds to minimize bias due to electrostatic charge.
Multiple weight measurements were conducted before and after sampling, with the difference
between weighings being less than 10 µg for each condition, respectively. The difference
between substrate weights before and after sampling was equated to total gravimetric mass.
Black carbon was measured using a Multi-wavelength Absorption Black Carbon Instrument
(MABI; Australian Nuclear Science and Technology Organisation). The MABI optically
quantifies black carbon concentrations by detecting the absorption at seven wavelengths (405,
465, 525, 639, 870, 940, and 1050 nm); however, the wavelength at 870 nm is used here as black
carbon is the primary absorber at that wavelength (Cruz et al., 2019; Ramachandran and Rajesh,
2007; Ran et al., 2016).
Meteorological parameters were measured at MO during the study period using a Davis Vantage
Pro2$^{TM}$ Plus automatic weather station, which was located on the roof of MO. Measured
parameters of relevance included temperature, accumulated rain, RH, and solar radiation. Data
were collected in five-minute increments and were cleaned based on the method of Bañares et al.
(2018) to verify values were in acceptable ranges. The meteorological parameters, except for
rain, were averaged over each sampling period while rain was summed over time to obtain the
accumulated precipitation for a sampling period. There were two periods where the automatic
weather station located at MO had missing values, 6 November – 27 November 2018 and 7
August – 3 September 2019. In these cases, missing values were substituted with values from a
secondary automatic weather station located approximately 2 km away (14.63° N, 121.06° E),
and if missing data still persisted, a tertiary station located 5 km away (14.67° N, 121.11° E) was
used. Identical data cleaning procedures were implemented for the secondary and tertiary sites.





2.3 Concentration weighted trajectories (CWT)
A CWT analysis was conducted to identify sources of detected species. The method assigns a
weighted concentration to a grid that is calculated by finding the mean of sample concentrations
that have trajectories crossing a particular cell in the grid (e.g., Dimitriou, 2015; Dimitriou et al.,
2015; Hilario et al., 2020a; Hsu et al., 2003). The software TrajStat (Wang et al., 2009)
determines CWT profiles by using back-trajectories from the NOAA Hybrid Single-Particle
Lagrangian Integrated Trajectory (HYSPLIT) model (Rolph et al., 2017; Stein et al., 2015).
Three-day back-trajectories were obtained with an ending altitude of 500 m above ground level
using the Global Data Assimilation System (GDAS) and the "Model vertical velocity" method.
The choice of 500 m is based on representativeness of the mixed layer and having been widely
used in other studies (e.g., Crosbie et al., 2014; Mora et al., 2017; Sorooshian et al., 2011).
Trajectories were obtained every 6 hours after MOUDI sampling began for each sample set,
yielding approximately nine trajectories per set. A grid domain of 95° to 150° E longitude and -
5° to 45° N latitude was used with a grid cell resolution of 0.5° × 0.5°. The analysis was
performed for each measured organic acid and MSA for the full diameter range of MOUDI sets
(0.056 – 18 μm). A weighting function was applied to the CWT plots to minimize uncertainty;
hereafter CWT plots will be referred as WCWT plots.

2.4 Positive matrix factorization (PMF)
PMF analysis was applied to identify sources and their relative importance for the mass
concentration budgets of the species discussed in this work. Model simulations were conducted
based on MOUDI data for the diameter range of 0.056 – 18 μm. Nineteen species (Al, Ti, K, Rb,
V, Ni, As, Cd, Pb, $Na^+$, $Cl^-$, $NO_3^-$, $SO_4^{2-}$, MSA, adipate, succinate, maleate, oxalate, and
phthalate) were included in the analysis and categorized as "strong". Each individual stage of
MOUDI sets was considered an independent variable for the analysis. Missing values or values
below detection limit were replaced with zeros with the exception of sets where ICP-QQQ
analysis was not performed (57, 59, 60, 61, 62, 64, 65). Those missing values were replaced with
the geometric mean for each respective stage. The uncertainty for each stage and species was
calculated as follows:
$Uncertainty = 0.05 * [x] + LOD$ (Eq. 1)
where [x] is the concentration of the species (Reff et al., 2007). No additional uncertainty was
added to account for any unconsidered errors for all species. The uncertainty of the model output
was evaluated using displacement (DISP), bootstrapping (BS), and bootstrapping with
displacement (BS-DISP). For BS, 100 resamples were used and a value of 0.6 was used as a
threshold for the correlation coefficient (r) to pass as successful mapping for each simulation.
To qualify as a valid result, reported PMF results had to meet the following criteria: (i) factors
mapped with BS runs, (ii) no factor swaps in DISP, (iii) dQ values being close or equal to 0%,
and (iv) no factor swaps in BS-DISP where Al, Ti, K, Rb, V, Ni, As, Cd, Pb, $Na^+$, $Cl^-$, $NO_3^-$, and



SO$_4^{2-}$ were displaced. PMF diagnostics can be seen in Table S4 based on the method of Brown et
al. (2015).

**3. Results**
A brief overview of the species being examined is first provided before reviewing concentration
statistics. MSA is an oxidation product of dimethylsulfide (DMS) emitted primarily from the
ocean (Berresheim, 1987; Saltzman et al., 1983), but it can also be formed from dimethyl
sulphoxide (DMSO) emitted from anthropogenic sources such as industrial waste (Yuan et al.,
2004). Gaseous MSA can become associated with particulate matter via new particle formation
(Dawson et al., 2012), and through heterogeneous reactions or condensation onto existing
particles (De Bruyn et al., 1994; Hanson, 2005). Of the three saturated dicarboxylic acids,
succinate (C$_4$) and adipate (C$_6$) are larger chain dicarboxylic acids linked to ozonolysis of cyclic
alkenes, which is common in areas with extensive vehicular emissions (Grosjean et al., 1978;
Hatakeyama et al., 1987). They can also be emitted via processes such as meat cooking (Rogge
et al., 1993) and biomass burning (Kawamura et al., 2013; Pereira et al., 1982) and can be
secondarily formed by the photo-oxidation of higher chain organic acids, such as azelaic acid
(Bikkina et al., 2014; Ervens et al., 2004). Oxalate (C$_2$) is the smallest of those three acids and is
usually the most abundant on a mass basis of all dicarboxylic acids in tropospheric aerosols as it
represents an end-product in the oxidation of both larger-chain carboxylic acids and also
glyoxylic acid (Ervens et al., 2004). It can be emitted via direct emissions such as from biomass
burning (Graham et al., 2002; Narukawa et al., 1999; Xu et al., 2020), combustion exhaust
(Kawamura and Kaplan, 1987; Kawamura and Yasui, 2005; Wang et al., 2010), and from
various biogenic sources (Kawamura and Kaplan, 1987). Maleate (C$_4$) is an unsaturated
dicarboxylic acid originating from combustion engines, including via direct emissions
(Kawamura and Kaplan, 1987) and secondarily produced from the photo-oxidation of benzene
(Rogge et al., 1993). Lastly, phthalate (C$_8$) represents an aromatic dicarboxylic acid associated
with incomplete combustion of vehicular emissions (Kawamura and Kaplan, 1987) and oxidation
of naphthalene or other polycyclic aromatic hydrocarbons (Fine et al., 2004; Kawamura and
Ikushima, 2002; Kawamura and Yasui, 2005). However, it has also been linked to biomass
burning (Kumar et al., 2015) and burning of plastic material such as polyvinyl chloride (PVC)
products, garbage, and plastic bags (Agarwal et al., 2020; Claeys et al., 2012; Fu et al., 2012; Li
et al., 2019; Nguyen et al., 2016; Simoneit et al., 2005). Secondary formation via aqueous-phase
chemistry has been documented for these organic acids (Kunwar et al., 2019; Sorooshian et al.,
2007a; Sorooshian et al., 2010; Sorooshian et al., 2006; Wonaschuetz et al., 2012) and MSA
(Hoffmann et al., 2016).
Meteorological data are next summarized based on average values temporally coincident with
each MOUDI sample set period for each of the seasons. The exception to this was the
accumulated rainfall, which was summed for the MOUDI set duration. Temperatures were stable
during the different seasons: 28.0 ± 1.04 °C (SWM18), 28.9 ± 0.8 °C (Trans), 28.3 ± 1.9 °C
(NEM), and 28.4 ± 1.5 °C (SWM19). Solar radiation was the highest during the Trans (279.61 ±
19.68 W m$^{-2}$) and NEM (304.01 ± 67.54 W m$^{-2}$) seasons, and lowest during the SWM18 (225.32



$\pm$ 56.26 W m$^{-2}$) and SWM19 (256.05 $\pm$ 86.88 W m$^{-2}$) seasons owing largely to more cloud cover.
Accumulated rain was highest for both SWM seasons (SWM18: 29.78 $\pm$ 27.28 mm; SWM19:
16.66 $\pm$ 23.98 mm) and much lower during the Trans (1.00 $\pm$ 1.11 mm) and NEM (2.20 $\pm$ 6.70
mm) seasons. Relative humidity was relatively consistent across seasons: SWM18 (69.6 $\pm$ 5.0
%), Trans (69.2 $\pm$ 2.2 %), NEM season (62.4 $\pm$ 8.0 %), SWM19 (72.6 $\pm$ 11.7 %). Finally, Fig. 1
summarizes predominant wind patterns for each season based on HYSPLIT back-trajectories
collected every 6 hours during sampling periods. The SWM18 and SWM19 seasons were
characterized by predominantly southwesterly winds, while the NEM and Trans seasons
experienced mostly northeasterly winds. In conclusion, there was much higher potential for wet
scavenging during the SWM seasons, with the potential for more photochemical reactivity in the
NEM and Trans seasons owing to enhanced incident solar radiation. As humidity was generally
enhanced year-round, there was the likelihood of aqueous-phase processing to occur in all
seasons. The combination of sustained RH, low boundary layer height, and high surface-level
particle concentrations have been suggested to counteract the effects of wet deposition on total
particle concentration in Metro Manila (Hilario et al., 2020a).

3.1 Bulk aerosol measurements
The range, mean, and standard deviation of concentrations integrated across the MOUDI
diameter range (0.056 – 18 μm) are shown in Table 1 for each organic acid and MSA for all
seasons. In order of decreasing concentration, the following was the order of abundance based on
the cumulative dataset: oxalate (148.59 $\pm$ 94.26 ng m$^{-3}$) > succinate (9.53 $\pm$ 22.25 ng m$^{-3}$) >
maleate (9.52 $\pm$ 19.66 ng m$^{-3}$) > phthalate (8.68 $\pm$ 13.77 ng m$^{-3}$) > adipate (7.60 $\pm$ 9.38 ng m$^{-3}$) >
MSA (5.40 $\pm$ 5.23 ng m$^{-3}$). The relative order of abundance varies for the sub- and super-
micrometer ranges with the only consistent feature being that oxalate was the most abundant
species. This result was consistent with past works showing oxalate to be the most abundant
organic acid in different global regions (e.g., Decesari et al. (2006); Kerminen et al. (1999);
Sorooshian et al. (2007b); Ziemba et al. (2011)).
Figure 2 shows the combined contribution of the organic acids and MSA to total gravimetric
mass, while Table S5 summarizes percent contributions of individual species to total mass for
different size bins. Combined, the measured organic acids and MSA accounted for only a small
part of the total cumulative mass (0.80 $\pm$ 0.66 %), ranging from 0.23 – 1.49 % across the 11
individual gravimetric sets. When the combined contribution of organic acids and MSA to total
gravimetric mass were separated by season, results are generally the same (Fig. S1), with
differences in the percent range being as follows: SWM18 = 0.64 %; Trans = 0.95 %; NEM =
0.50 – 1.49 %; and SWM19 = 0.23 – 0.83 %. The highest contribution of these organic acids and
MSA occurred for MOUDI sets collected 12 – 14 March 2019 during the NEM season, which
accounted for 1.49 % (0.50 μg m$^{-3}$) of the total mass. The lowest contribution of these organic
acids and MSA occurred for MOUDI sets collected 11 – 13 September 2019 during the SWM19
season, which accounted for 0.23 % (0.06 μg m$^{-3}$) of the total mass. The summed contributions
of the six species were nearly the same in the sub- and supermicrometer ranges (0.78 $\pm$ 0.74 %
and 0.84 $\pm$ 0.58 %, respectively). Their contributions peaked in the two sizes bins covering the





range between 0.56 and 1.8 μm (0.56 – 1 μm: 1.06 ± 1.01 %; 1 – 1.8 μm: 1.01 ± 0.78 %). After
accounting for all measured species (BC, water-soluble species), there still remained 33.74 ±
19.89 % (range: 23.86 – 50.88 %) of unresolved mass. Therefore, the six species of interest in
this work only explain a small amount of the region's mass concentrations and further work is
still needed to resolve the remaining components, which presumably is dominated by water-
insoluble organics and elements. Of most need is to resolve those missing components in the
supermicrometer range, where Table S5 shows that the unresolved fraction is 69.10 ± 25.91 %,
in contrast to 17.78 ± 17.25 % for the submicrometer range.
Although there are fairly wide ranges in concentration for the individual species, a few features
are noteworthy based on the cumulative dataset. First, the oxalate concentrations are lower than
expected for such a highly polluted area, as will be expanded upon in Sect. 4.6. Second, there is a
significant decrease in concentration after oxalate for the remaining five species, which had
similar mean concentrations. Lastly, although the sampling site is on an island and close to
marine sources, MSA is surprisingly the least abundant among the six species of interest.
Mean mass concentrations of these species varied greatly by season as visually shown in Fig. 3a
and summarized numerically in Table 1. In contrast, Fig. 3b shows that the mass fractions of the
six species did not change much seasonally owing to the dominance of oxalate (37.67 – 472.82
ng m$^{-3}$), which accounted for between 69.1-87.3 % of the cumulative concentration of the six
species across the four seasons. Important features with regard to seasonal mass concentration
differences include the following: (i) maleate concentrations were much higher in the SWM18
and SWM19 seasons; (ii) the lowest overall concentrations of most species, besides oxalate and
succinate (lowest in SWM19), were observed in the NEM season; (iii) oxalate and phthalate
were the only species that peaked in the Trans period, whereas the rest of the species peaked in
either SWM18 or SWM19; and (iv) succinate and phthalate were peculiarly much more
enhanced in SWM18 than SWM19, pointing to significant variability between consecutive years.

3.2 Source apportionment
To help elucidate how different emissions sources impact the six species, PMF analysis was
conducted and yielded a solution with five source factors using year-round data (Fig. 4). The five
sources are as follows in decreasing order of their contribution to the total mass based on the sum
of species used in the PMF analysis (Fig. 4): combustion (32.1 %), biomass burning (20.9 %),
sea salt (20.9 %), crustal (14.2 %), and waste processing (11.9 %). The contribution of each
source to the total concentration of organic acids and MSA was as follows: combustion (33.5 %),
biomass burning (29.0 %), crustal (27.0 %), waste processing (9.8 %), and sea salt (0.6 %). The
source factor names were determined based on the enhancement of the following species (Fig.
4): crustal (Al, Ti) (Harrison et al., 2011; Malm et al., 1994; Singh et al., 2002), biomass burning
(K, Rb) (Andreae, 1983; Artaxo et al., 1994; Braun et al., 2020; Chow et al., 2004; Echalar et al.,
1995; Ma et al., 2019; Schlosser et al., 2017; Thepnuan et al., 2019; Yamasoe et al., 2000), sea
salt (Na, Cl) (Seinfeld and Pandis, 2016), combustion (V, Ni, As) (Allen et al., 2001; Linak et al.,
2000; Mahowald et al., 2008; Mooibroek et al., 2011; Prabhakar et al., 2014; Wasson et al.,



2005), and waste processing (Cd, Pb) (Cruz et al., 2019; Gullett et al., 2007; Iijima et al., 2007;
Pabroa et al., 2011). While both $SO_4^{2-}$ and $NO_3^-$ are secondarily produced, the latter is more
commonly linked to supermicrometer particles (Allen et al., 1996; Dasgupta et al., 2007;
Fitzgerald, 1991; Maudlin et al., 2015), including in the study region (Cruz et al., 2019).
Additionally, Al, K, and Cl are linked to biomass burning (Reid et al., 1998; Reid et al., 2005;
Schlosser et al., 2017; Wonaschütz et al., 2011). The source factor names should be interpreted
with caution, as a single profile may consist of a mix of sources (e.g., waste processing). It
should be noted that Cruz et al. (2019) performed PMF analysis for only the SWM18 season,
which yielded similar and additional sources for only the SWM18 season, whereas this study
used year-round data.
To provide size-resolved context for the five aerosol sources, Fig. 5 shows their respective
reconstructed mass size distributions based on PMF output. Distributions for combustion,
biomass burning, and waste processing primarily peaked in the submicrometer range, while
crustal and sea salt sources primarily peaked in the supermicrometer range. Combustion and
biomass burning factors showed a dominant peak between 0.32 – 0.56 μm, whereas waste
processing had a peak between 0.56 – 1 μm. The crustal and sea salt factors exhibited their peak
concentrations between 1.8 – 5.6 μm. Both crustal and biomass burning sources showed signs of
bimodal size distributions with a minor peak in the sub- and supermicrometer ranges,
respectively.
As reported in Table 2, combustion was the largest contributor to the cumulative mass
concentrations of organic acids and MSA, with the largest influence being for maleate (69.7 %)
and MSA (57.4 %). Biomass burning was marked by its significant contribution to succinate
(90.3 %). The sea salt source showed minor contributions to phthalate (9.9 %) and adipate (4.7
%). The crustal source contributed appreciably to adipate (35.9 %) and oxalate (31.2 %), with the
rest of the organic acid or MSA species being less influenced (0.1 – 13.3 %). Organic acids have
been shown in past work to be associated with mineral dust (Russell et al., 2002), including both
oxalic and adipic acids (Falkovich et al., 2004; Kawamura et al., 2013; Sullivan and Prather,
2007; Tsai et al., 2014), although less has been documented for adipate. Wang et al. (2017) and
Yao et al. (2003) both report that gaseous acids are likely to adsorb onto supermicrometer
particles that are highly alkaline, such as dust. The waste processing factor contributed to
maleate (30.1 %), oxalate (10.5 %), and MSA (1.4 %). An unexpected result was that the sea salt
factor did not contribute to MSA even though the latter is derived from ocean-emitted DMS; the
results of Table 2 suggest that other sources such as biomass burning and industrial activities are
more influential in the study region similar to other regions like Beijing (Yuan et al., 2004) and
coastal and inland areas of California (Sorooshian et al., 2015).

3.3 Species interrelationships
Correlation analysis was conducted for the same species used in the PMF analysis to quantify
interrelationships and to gain additional insight into common production pathways. Correlation
coefficients (r) values are reported in Table 3 for for the sub- and supermicrometer ranges,





whereas results for full size range are shown in Table S6. Values are only shown and discussed
subsequently for correlations with p-values below 0.05. Unless otherwise stated, correlations
discussed below correspond to the full size range for simplicity, whereas notable results when
contrasting the two size ranges ($< 1$ μm and $> 1$ μm) are explicitly mentioned.
MSA exhibited a statistically significant correlation with Rb ($r = 0.37$), suggestive of its link
with biomass burning as Rb has been shown in the study region to be a biomass burning marker
(Braun et al., 2020). Additionally, MSA was correlated with Na, $NO_3^-$, and $SO_4^{2-}$ (r: 0.35 – 0.59),
which are associated with marine aerosol (e.g., sea salt, DMS, shipping) but also biomass
burning. The supermicrometer results indicate MSA was correlated only with Na ($r = 0.32$), due
presumably to co-emission from both crustal and sea salt sources, with the former commonly
linked to biomass burning (Schlosser et al., 2017). For the submicrometer range, MSA was
correlated with Rb and $SO_4^{2-}$ (r: 0.39 – 0.60), which are derived from biomass burning and other
forms of combustion, consistent with smaller particles formed secondarily from gas-to-particle
conversion processes. That is also why MSA was well correlated with succinate, oxalate, and
phthalate (r: 0.53 – 0.67), which were also prominent species in either (or both of) the biomass
burning and combustion factors.
Adipate only exhibited significant correlations with maleate and phthalate for the full diameter
range (r: 0.43 – 0.45), while maleate was correlated only with adipate. In contrast, succinate,
oxalate, and phthalate were correlated with a wide suite of species, indicating that maleate and
adipate exhibited more unique behavior in terms of their production routes. Succinate, oxalate,
and phthalate similarly exhibited significant correlations with each other, and species linked to
crustal sources (Al, Ti, Na), sea salt (Na), and biomass burning (Rb). Succinate and oxalate in
particular were better correlated with tracer species related to either dust or sea salt (Al, Na) in
the supermicrometer range, and were correlated with each other also in that size range.

3.4 Cumulative size distribution variations
Mass size distributions for each individual organic acid and MSA are shown for the full study
period in Fig. S2 and seasonal mass size distributions can be seen in Figs. 6-11. General
information for the cumulative dataset will be described here before examining seasonal results
in Sect. 4. While significant variability exists between individual sets for the cumulative dataset,
a few general features are evident: (i) mass size distributions all appear multi-modal with the
exception of maleate, which on average exhibited a uni-modal profile; (ii) all species show a
larger peak in the submicrometer range versus supermicrometer sizes; (iii) phthalate and adipate
show more comparable peaks in the sub- and supermicrometer range; and (iv) the size bin where
the peaks occur vary between species. These results point to differences in the species with
regard to their source, formation mechanism, and eventual fate.
One factor relevant to the mass size distribution plots is the source origin of sampled air masses.
The WCWT plots in Fig. 12 reveal the bulk of the concentration of a few species (e.g., phthalate,
succinate, and MSA) was explained by southwesterly flow. Consistent with the PMF results
showing that the biomass burning factor contributed the most to these three species, the





predominant fire sources were to the southwest of Luzon. Past work has linked these areas to
significant biomass burning influence over Luzon and the South China Sea during the SWM
season (Atwood et al., 2017; Ge et al., 2017; Hilario et al., 2020b; Reid et al., 2016; Song et al.,
2018; Wang et al., 2013; Xian et al., 2013). Noteworthy is that the WCWT maps for SWM18
reveal more influence from the biomass burning hotspots to the southwest (e.g., Borneo and
Sumatra), in contrast to SWM19, pointing to more biomass burning influence in the former
season. Oxalate's WCWT profile shows the most spatial heterogeneity in terms of source
regions; this is consistent with it being an end-product in the oxidation of other carboxylic acids
that can originate from numerous sources. Finally, adipate and maleate similarly showed a
localized hotspot in terms of where their greatest influence originated, approximately 290 km to
the north-northwest of MO. This could be partly linked to the Sual coal-fired power station
located near that area where an ash disposal site is also in close proximity. The uniquely similar
WCWT maps between adipate and maleate is consistent with them having few correlations, if
any, with species aside from each other (Table S6). Subsequent sections discuss each organic
acid and MSA in more detail including seasonal behavior.

**4. Discussion**
4.1 Phthalate
Results from Sect. 3 show that phthalate has the following characteristics: (i) influenced most by
biomass burning (49.5 %), followed by combustion (27.4 %), crustal sources (13.3 %), and then
sea salt (9.9 %); (ii) significant correlations with more species in Table S6 than any other organic
acid or MSA; (iii) comparable mass size distribution modes in the sub- and supermicrometer size
ranges; (iv) highest mass concentration in the Trans period, but also exhibited significantly
different concentrations between the two SWM seasons; and (v) had concentrations dominated
by sources to the southwest. Previous studies measuring phthalate in other regions have found
concentrations of 40.1 – 105 ng m$^{-3}$ (Hong Kong; PM$_{2.5}$; Ho et al. (2006)), <0.01 – 7.6 ng m$^{-3}$
(remote marine; total suspended particles (TSP); Kawamura and Sakaguchi (1999)), 0.16 – 3.25
ng m$^{-3}$ (Arctic; TSP; Kawamura et al. (2010)), and 0 – 57.3 ng m$^{-3}$ (Rondônia, Brazil; PM$_{2.5}$;
Decesari et al. (2006)). The latter was more consistent with concentrations in this study (0 –
67.02 ng m$^{-3}$), albeit the size ranges examined vary. A more detailed examination based on
seasonally resolved mass size distributions and WCWT maps follows to try to gain more insights
into this species. Although not referenced hereafter, Table S7 provides numerical details about
mass concentration mode sizes and associated concentrations for each season and the cumulative
dataset for each species.
The average size distributions for phthalate appeared bi-modal for each individual season (Fig.
6). Depending on the season, concentration peaks occurred in three separate MOUDI stages for
the submicrometer range, and between 1.8 – 3.2 or 3.2 – 5.6 μm in the supermicrometer range.
The NEM season was unique in that the supermicrometer peak was considerably more
pronounced than in the submicrometer range, which was a rare occurrence in this study for all
species except adipate. Phthalate appears in the submicrometer range due to secondary formation



by photo-oxidation (i.e., Kautzman et al., 2010; Kawamura and Ikushima, 2002; Kawamura and
Yasui, 2005; Kleindienst et al., 2012) and from primary emissions (i.e., combustion,
biomass/waste burning) (i.e., Deshmukh et al., 2016; Kawamura and Kaplan, 1987; Kumar et al.,
2015; Kundu et al., 2010). Its general presence in the supermicrometer range, especially during
the NEM season, can be explained by possible adsorption onto larger particles such as dust and
sea salt (i.e., Wang et al., 2012; Wang et al., 2017). Others have observed an enhancement in
phthalate in the supermicrometer mode, specifically in Xi'an, China, due to suspected adsorption
of its vapor form (Wang et al., 2012) derived from photo-oxidation of naphthalene (Ho et al.,
2006; Wang et al., 2011; Wang et al., 2012; Wang et al., 2017).
WCWT results for phthalate (Fig. S3) showed high concentrations across all seasons coming
from the southwest, most notably in the SWM18 and SWM19 seasons. The significant reduction
in phthalate levels from SWM18 ($16.75 \pm 24.80$ ng m$^{-3}$) to SWM19 ($5.72 \pm 7.41$ ng m$^{-3}$) is
coincident with stronger influence from biomass burning from the southwest in 2018. Figure 3
showed that the highest concentration of phthalate occurred in the Trans period, assumed to be
largely due to local emissions (e.g., vehicular traffic) based on the WCWT results with
significant influence in the immediate vicinity of Luzon unlike the other seasons. The peculiar
size distribution results for the NEM season can be explained by the WCWT map showing
strong influence from the northeast, which likely includes supermicrometer aerosol influences
from sea salt and dust from East Asia. The reduced influence of upwind anthropogenic and
biomass burning emissions during the NEM season can explain the lower seasonal
concentrations, especially in the submicrometer size range (Hsu et al., 2009).

**4.2 Adipate**
Adipate was shown in Sect. 3 to have the following features: (i) influenced most by crustal
sources (35.9 %), followed by combustion (32.9 %), biomass burning (26.4 %), and finally sea
salt (4.7 %); (ii) only correlated with maleate and phthalate; (iii) comparable concentrations in
the sub- and supermicrometer size ranges, with a mode between 5.6 and 10 µm; (iv) highest mass
concentration in the SWM seasons, but especially the SWM19 season; and (v) concentrations
dominated by sources from the southwest as well as from the northwest. Concentrations for
adipate measured in other regions include $3.78 – 32.1$ ng m$^{-3}$ (Hong Kong; PM$_{2.5}$; Ho et al.
(2006)), $3.8 – 16.8$ ng m$^{-3}$ (Rondônia, Brazil; PM$_{2.5}$; Decesari et al. (2006)), $0.60 – 13$ ng m$^{-3}$
(remote marine; TSP; Kawamura and Sakaguchi (1999)) and $0.21 – 2.94$ ng m$^{-3}$ (Arctic; TSP;
Kawamura et al. (2010)). The range in this study was $0 – 43.83$ ng m$^{-3}$, with an upper bound that
exceeded those in the previous works.
Mass size distributions for adipate were the most variable in structure compared to the other five
species with multiple peaks present at different sizes (Fig. S2). In general, its distributions
appeared uniquely and consistently tri-modal with the exception of the SWM18 season where it
was bi-modal (Fig. 7). Modes appeared between $0.10 – 0.18$ µm and $0.32 – 0.56$ µm for the
submicrometer range, and between $1.0 – 1.8$ µm and $3.2 – 5.6$ µm in the supermicrometer range.
The SWM19 season was unique for adipate as the highest peak was in the supermicrometer





range and it was higher than any other peak across the other seasons. Submicrometer adipate is
likely derived from a photo-oxidation of higher chain organic acids (i.e., van Drooge and
Grimalt, 2015), ozonolysis of vehicular emissions (i.e., Grosjean et al., 1978), and from the
primary emissions of biomass burning (i.e., Graham et al., 2002). The appearance in the
supermicrometer range likely due to adsorption onto larger particles such as dust and sea salt
(e.g., Wang et al., 2012; Wang et al., 2017). As the PMF results suggest crustal sources were
more influential for adipate in contrast to sea salt, dust was more likely the supermicrometer
particle type that adipate preferentially partitioned to. The source of the dust was likely a
combination of long-range transport from (i) the southwest especially during biomass burning
periods, (ii) East Asia, and (iii) locally generated dust via anthropogenic activities (Fig. S4).
Past work in the study region showed that broad mass size distributions with comparable
concentrations in the sub- and supermicrometer ranges were coincident with wet scavenging
(Braun et al., 2020) and appreciable primary emissions of sea salt and dust (AzadiAghdam et al.,
2019; Cruz et al., 2019). Scavenging was suggested to remove transported pollution while
allowing for more pronounced contributions from more localized emissions, which could include
vehicular traffic, sea salt, and anthropogenic forms of dust (e.g., road dust, construction), all of
which are consistent with adipate's mass size distribution data and WCWT maps (Fig. S4)
showing high concentrations predominately around Luzon for all seasons.

### 4.3 Succinate
Succinate exhibited the following characteristics: (i) influenced primarily by biomass burning
(90.3 %) followed by crustal sources (9.7 %); (ii) exhibited high correlation coefficients (0.67 –
0.76) with oxalate, phthalate, and MSA (Table S6); (iii) mass was focused in the submicrometer
range; (iv) highest mass concentrations were in the SWM18 season, and, similar to phthalate,
showed a significant reduction in the SWM19 season; and (v) had concentrations dominated by
sources from the southwest. The range of concentrations in this study ($0 - 166.28$ ng m$^{-3}$) is
somewhat consistent with those from other regions: $61.8 - 261$ ng m$^{-3}$ (Rondônia, Brazil; PM$_{2.5}$;
Decesari et al. (2006)), $13.1 - 121$ ng m$^{-3}$ (Hong Kong; PM$_{2.5}$; Ho et al. (2006)), $9.2 - 31.7$ ng m$^{-3}$
$^{3}$ (New England, USA; $0.4 - 10$ µm; Ziemba et al. (2011)), $0.29 - 16$ ng m$^{-3}$ (Remote Marine;
TSP; Kawamura and Sakaguchi (1999)), and $1.35 - 12.9$ ng m$^{-3}$ (Arctic; TSP; Kawamura et al.
561 (2010)).

The average size distributions for succinate varied in the number of peaks present ($2 - 4$), but on
average were bi-modal with a submicrometer mode usually between $0.32 - 0.56$ µm or $0.56 - 1.0$
µm, and a smaller supermicrometer mode between either $1.8 - 3.2$ µm or $3.2 - 5.6$ µm (Fig. 8).
The chief source of succinate, which is concentrated in the submicrometer peak, is biomass
burning (Pratt et al., 2011; Vasconcellos et al., 2010), which is reinforced by the PMF results
(Table 2), its high correlation with the biomass burning tracer Rb ($r = 0.67$; Table S6) (Braun et
al., 2020) and WCWT maps showing its most pronounced influence from biomass burning
hotspots to the southwest during the SWM18 season (Fig. S5). There likely was also local
biomass burning during the NEM season contributing to succinate concentrations. Hilario et al.





(2020a) showed based on satellite data that local fire activity peaks between March and May.
There was less influence from biomass burning in the SWM19 season, which is why succinate's
levels were lower ($4.73 \pm 7.43$ ng m$^{-3}$) than in the SWM18 season ($21.61 \pm 43.10$ ng m$^{-3}$).
Similar to phthalate and adipate, there were more local hotspots of concentration in seasonal
WCWT maps pointing to local anthropogenic sources such as vehicular traffic and the presence
of supermicrometer particles like dust and sea salt that succinate can partition to (e.g., Wang et
al., 2012; Wang et al., 2017).

4.4 Maleate
The results of Sect. 3 showed that maleate had the following attributes: (i) influenced most by
combustion (69.7 %), followed by waste processing (30.1 %), and then barely by crustal sources
(0.2 %); (ii) only correlated with adipate of all species shown in Table S6; (iii) showed a uni-
modal mass size distribution, with negligible contribution in the supermicrometer range; (iv)
highest mass concentration in the SWM19 season, but was comparable to the SWM18 season;
and (v) WCWT maps showed the most localized sources as compared to the other species
examined (Fig. 11). Maleate concentrations have been reported for other regions as follows: 7 –
75 ng m$^{-3}$ (Rondônia, Brazil; PM$_{2.5}$; Decesari et al. (2006)), 2.21 – 37.2 ng m$^{-3}$ (Hong Kong;
PM$_{2.5}$; Ho et al. (2006)), 4.9 – 9.2 ng m$^{-3}$ (New England, USA; 0.4 – 10 μm; Ziemba et al.
(2011)), 0.04 – 3.8 ng m$^{-3}$ (remote marine; TSP; Kawamura and Sakaguchi (1999)), and 0.04 –
0.83 ng m$^{-3}$ (Arctic; TSP; Kawamura et al. (2010)). The values reported for this study region
tended to be higher (0-119.19 ng m$^{-3}$), which is unsurprising as vehicular emissions are so
prominent in the Metro Manila region (Alas et al., 2018; Kecorius et al., 2017).
The average seasonal size distributions for maleate appeared to be uni-modal with peaks between
0.32 – 0.56 μm and 0.56 – 1.0 μm (Fig. 9). The absence of a supermicrometer peak, in contrast to
most other species, indicates that it had less diverse sources and was derived from combustion
emissions without being adsorbed onto supermicrometer particles like the other species
investigated. The association of maleate with the waste processing source factor in Table 2 can
be explained partly by the burning and recycling of electronic waste (Cruz et al., 2019; Gullett et
al., 2007; Iijima et al., 2007). The Pabroa et al. (2011) study reported that there are few licensed
operators for battery recycling, but there are numerous unregulated melters frequently melting
metal and discarding the waste.
Seasonal WCWT maps for maleate (Fig. S6) consistently showed hotspots around Luzon
indicative of local emissions. Maleate concentrations for the SWM18 ($18.68 \pm 14.89$ ng m$^{-3}$) and
SWM19 ($19.44 \pm 34.04$ ng m$^{-3}$) were significantly higher than the other seasons (Trans: $3.81 \pm$
$4.23$ ng m$^{-3}$; NEM: $1.65 \pm 3.65$ ng m$^{-3}$), and this could likely be due to increased traffic
emissions because of gridlock due to intense rainfall. It should be noted that the Ateneo de
Manila campus has student break periods in March, April, May, and December (Hilario et al.,
2020a); those months pertain to the NEM season, which could lead to lower combustion
emissions from vehicles (e.g., maleate and phthalate). Although the SWM season is associated
with enhanced precipitation over Metro Manila, lower boundary layer height and appreciable RH





values could counteract wet scavenging to some degree by promoting aqueous processing of
aerosol (Hilario et al., 2020a). Furthermore, maleate's largely submicrometer size distribution
(Fig. 9) may reduce the efficiency of wet scavenging (Greenfield, 1957).

4.5 Oxalate
Oxalate was shown to have the following traits: (i) influenced somewhat uniformly by
combustion (32.9 %) and crustal (31.2 %) sources, followed by biomass burning (25.4 %), and
waste processing (10.5 %); (ii) only organic acid to correlate with combustion tracers (V, Ni);
(iii) pronounced presence in both the sub- and supermicrometer size ranges; (iv) highest mass
concentrations in the Trans period; and (v) had contributions from the southwest, east/northeast,
and locally. Oxalate concentrations in this study (148.59 ± 94.26 ng m$^{-3}$) were surprisingly low
for such a polluted megacity with strong regional sources. For context, concentrations in a few
other regions are as follows: 1.14 µg m$^{-3}$ in Sao Paulo, Brazil (Souza et al., 1999); 0.27 – 1.35 µg
m$^{-3}$ in Tokyo, Japan (Kawamura and Ikushima, 2002; Sempére and Kawamura, 1994); 0.49 µg
m$^{-3}$ in Los Angeles, California (Kawamura et al., 1985); 220 – 300 ng m$^{-3}$ in Nanjing, China
(Yang et al., 2005); 75 – 210 ng m$^{-3}$ for multiple sites in Europe (Hungary, Belgium, Finland)
(Maenhaut et al., 2011); 12.3 – 33.7 ng m$^{-3}$ in Cape San Juan, Puerto Rico (Jusino-Atresino et
al., 2016); 20 – 400 ng m$^{-3}$ in rural/urban Finland (Kerminen et al., 2000); and 1 – 42 ng m$^{-3}$
around the Atlantic Ocean/Antarctic (Virkkula et al., 2006).
The average size distributions for oxalate appeared bi-modal for each individual season with
modes between 0.32 – 0.56 µm and 0.56 – 1.0 µm for the submicrometer range and a separate
mode between 1.8 – 3.2 µm for the supermicrometer range (Fig. 10). A unique aspect for oxalate
was its consistency in having a bi-modal profile each season with the supermicrometer mode
always between 1.8 – 3.2 µm. Submicrometer oxalate likely originated from secondary
production from both biogenic and anthropogenic precursor emissions, and potentially from
primary emissions (i.e., combustion/biomass burning) (i.e., Decesari et al., 2006; Falkovich et
al., 2005; Golly et al., 2019; Kundu et al., 2010; Wang et al., 2010). Of all the six species
studied, oxalate was best correlated with $SO_4^{2-}$ (r = 0.69; Table S6), especially in the
submicrometer range (r = 0.72; Table 3), which is consistent with their common production
mechanism via aqueous processing (Sorooshian et al., 2006; Yu et al., 2005). Additionally, high
concentrations of oxalate in the Trans period indicate that photo-oxidation was an important
process for oxalate formation since the Trans period had low rain and high solar radiation. The
prominent supermicrometer presence was likely due to adsorption onto supermicrometer
particles. Past work by Sullivan and Prather (2007) reported the following with regard to
oxalate's behavior in coarse particles of relevance to this study: (i) oxalic acid was
predominately associated with mineral dust and to a lesser degree with aged sea salt; (ii) even
though most of the total mass was sea salt, there was more oxalate per mass of mineral dust than
sea salt; (iii) Asian dust particles are more alkaline as opposed to sea salt and therefore act as
better sinks for dicarboxylic acids than sea salt; and (iv) it is feasible that a large fraction of
supermicrometer dicarboxylic acid mass in remote marine air is associated with mineral dust and
not sea salt. The PMF results from the present study indicate that oxalate was much more



influenced by crustal sources (31.2 %) versus sea salt (0 %), similar to phthalate, adipate, and
succinate (Table 2). Reinforcing the relationship between oxalate and dust is the significant
correlation between oxalate and both Al (r = 0.59) and Ti (0.29) in the supermicrometer range.
WCWT results for oxalate (Fig. S7) showed high concentrations around Luzon for all seasons,
with the caveat that the SWM18 exhibited high concentrations coming from the southwest,
which has already been linked to biomass burning emissions. The difference in oxalate levels
between the SWM18 (177.86 $\pm$ 139.41 ng m$^{-3}$) and SWM19 (110.21 $\pm$ 62.06 ng m$^{-3}$) seasons is
largely due to the enhanced contribution of biomass burning in the former season since oxalate is
abundant in fire emissions (Falkovich et al., 2005; Mardi et al., 2018; Narukawa et al., 1999).

4.6 MSA
Previous sections revealed the following characteristics for MSA: (i) influenced most by
combustion (57.4 %), followed by biomass burning (41.2 %), waste processing (1.4 %), and then
crustal sources (0.1 %); (ii) significantly correlated with succinate, oxalate, phthalate, and $SO_4^{2-}$;
(iii) similar to maleate, primarily consisted of a submicrometer mass size distribution peak with
only minor contributions from the supermicrometer mode; (iv) concentration was highest during
the SWM18 season; and (v) had concentrations dominated by sources from the southwest.
Concentrations of MSA in this study were surprisingly low for a site so close to marine and
anthropogenic sources (0.10 – 23.23 ng m$^{-3}$). For context, MSA concentrations in other regions
are as follows: 30-60 ng m$^{-3}$ in Nanjing, China (Yang et al., 2005); 29 – 79 ng m$^{-3}$ for multiple
sites in Europe (Hungary, Belgium, Finland) (Maenhaut et al., 2011); 2.33 – 3.33 ng m$^{-3}$ in
Cape San Juan, Puerto Rico (Jusino-Atresino et al., 2016); 5 – 115 ng m$^{-3}$ in rural/urban Finland
(Kerminen et al., 2000); 2.8 – 20 ng m$^{-3}$ around the Atlantic Ocean/Antarctic (Virkkula et al.,
2006); ~7 ng m$^{-3}$ in Tucson, Arizona and ~101 ng m$^{-3}$ Marina, California (Sorooshian et al.,
2015); 29 – 66 ng m$^{-3}$ over the China Sea (Gao et al., 1996); 13 – 59 ng m$^{-3}$ at various coastal
and island sites over the North Pacific Ocean (Arimoto et al., 1996); and 34 $\pm$ 33 ng m$^{-3}$ over
Houston, Texas (Sorooshian et al., 2007b).
The average size distributions for MSA appeared uni-modal with the peak size being between
0.32 – 0.56 μm (Fig. 11). The consistent mass size distribution for MSA in all seasons, similar to
maleate, could be due to some combination of limited sources and production pathways.
Surprisingly, MSA showed no association to the sea salt source factor (Table 2) even though it
would be expected given that DMS is co-emitted from the ocean with sea salt. Instead,
combustion and biomass burning sources were more significant, which is consistent with some
past studies linking MSA to anthropogenic sources (Yuan et al., 2004) and biomass burning
(Sorooshian et al., 2015). WCWT results for MSA (Fig. S8) showed high concentrations coming
from the southwest during the SWM18 and SWM19 seasons, and from the east-northeast during
the NEM and Trans period.

**4. Conclusions**





This work used a 16-month long dataset of size-resolved aerosol composition to investigate the
nature of five organic acids (oxalate, succinate, adipate, maleate, and phthalate) and MSA in the
polluted Metro Manila region in the Philippines. Selected results are as follows in order of the
three major questions posed at the end of Sect. 1.
• Organic acids and MSA contribute only a small fraction to the total gravimetric aerosol
mass in Metro Manila ($0.80 \pm 0.66$ %). The combined contribution of these six species
was similar between the sub- and supermicrometer range (0.78 % and 0.84 %,
respectively). After accounting for water-soluble ions and elements, and black carbon,
there still was an unresolved mass fraction amounting to 33.74 % across all sizes, and
17.78 % and 69.10 % for sub- and supermicrometer sizes, respectively. Therefore, future
work is still warranted to identify what the missing fraction is comprised of, which is
speculated to be water-insoluble organics and elements.
• Oxalate was the most abundant of the six species accounting for 69.1 – 87.3 % of the
total combined mass of the six species depending on the season. However, the bulk
concentrations of oxalate were unusually low ($148.59 \pm 94.26$ ng m$^{-3}$) for such a polluted
area in contrast to other populated regions. Concentrations of the other five species were
much lower than oxalate, with mean levels for the entire study period being less than 10
ng m$^{-3}$. In particular, MSA exhibited the lowest mean concentration ($5.40 \pm 5.23$ ng m$^{-3}$).
It is unclear exactly as to the reason for the low concentrations of the examined species in
light of the diverse marine and anthropogenic sources in the region. The role of wet
scavenging, especially in the SWM seasons, will be the subject of future research.
• The six species exhibited different behavior seasonally, both in terms of relative
concentration and mass size distribution. The SWM18 season was uniquely different than
the SWM19 season, owing to more biomass burning emissions transported from the
southwest that yielded enhanced levels for most species in the submicrometer range,
especially succinate, MSA, oxalate, and phthalate. Enhanced precipitation in the SWM
seasons also was coincident with more influence from localized emissions leading to
enhanced levels in the sub- and supermicrometer ranges depending on the species. The
NEM season was characterized by generally lower concentrations of most species as air
was predominantly transported from the northeast with reduced influence of
anthropogenic and biomass burning emissions. Phthalate was enhanced in the
supermicrometer range during the NEM season due to presumed adsorption to Asian dust
and to a lesser extent sea salt. The Trans season was characterized by having strong
influence from localized emissions for all six species, which promoted especially high
concentrations for phthalate and oxalate in both the sub- and supermicrometer ranges.
• All species exhibited a prominent submicrometer peak that likely stemmed largely from
secondary formation from both anthropogenic and biogenic precursor emissions and was
especially prominent during the SWM18 season due to extensive biomass burning
influence. Biomass burning was an especially important source for succinate, phthalate,
MSA, oxalate, and adipate. All six species exhibited relatively low association with sea
salt particles; this was particularly interesting for MSA, which was instead better related
to combustion and biomass burning emissions. In contrast to sea salt, most species were



linked to crustal emissions as evident from peaks in the coarse mode during periods of
dust influence. Oxalate, adipate, phthalate, and succinate in particular preferentially
partitioned to dust rather than sea salt, potentially due to their affinity for alkaline particle
types. Oxalate was best correlated with sulfate, especially in the submicrometer mode,
explained by their common production via aqueous processing, which is common in the
study region owing to high humidity levels year-round.
The results of this study point to the importance of size-resolved measurements of organic and
sulfonic acids as this extensive dataset revealed important changes in mass size distributions
between species and for different seasons. The data point to the partitioning of these species to
coarse aerosol types and the potentially significant impact of precipitation on either the removal
or enhancement of species' mass size distribution modes; these topics warrant additional
research to put on firmer ground the sensitivity of these species to source regions, transport
pathway, and wet scavenging effects. More research is warranted to investigate the remaining
fraction of the unresolved mass (approximately one third of the gravimetric mass) that is not
accounted for by black carbon and the water-soluble constituents speciated in this work. This is
especially important for the supermicrometer range. Lastly, the current results point to the
question as to what drives the affinity of individual species towards the coarse mode for different
aerosol types (e.g., dust, sea salt), and how common this is for other regions.

## Data availability

Size-resolved aerosol data collected at Manila Observatory are described in Stahl et al. (2020b)
and archived on figshare (Stahl et al., 2020a) as well as on the NASA data repository at
DOI:10.5067/Suborbital/CAMP2EX2018/DATA001.

## Author contribution

MTC, MOC, JBS, RAB, ABM, CS, and AS designed the experiment. All coauthors carried out
various aspects of the data collection. MTC, RAB, CS, and AS conducted analysis and
interpretation of the data. CS and AS prepared the manuscript with contributions from the
coauthors.

## Competing interests

The authors declare that they have no conflict of interest.

## Acknowledgements

The authors acknowledge support from NASA grant 80NSSC18K0148 in support of the NASA
CAMP²Ex project. R. A. Braun acknowledges support from the ARCS Foundation. Cruz





acknowledges support from the Philippine Department of Science and Technology's ASTHRD
Program. A. B. MacDonald acknowledges support from the Mexican National Council for
Science and Technology (CONACYT). We acknowledge Agilent Technologies for their support
and Shane Snyder's laboratories for ICP-QQQ data.

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





**Table 1:** Seasonal concentrations (ng m$^{-3}$) of organic acids and MSA for all (0.056 – 18 μm), submicrometer (0.056 – 1 μm), and supermicrometer (1 – 18 μm) sizes measured in Metro Manila from July 2018 to October 2019. n = number of sets.

| Size/Species | | All (n = 54) Range | All (n = 54) Mean (SD) | SWM18 (n = 11) Range | SWM18 (n = 11) Mean (SD) | Trans (n = 3) Range | Trans (n = 3) Mean (SD) | NEM (n = 27) Range | NEM (n = 27) Mean (SD) | SWM19 (n = 13) Range | SWM19 (n = 13) Mean (SD) |
|---|---|---|---|---|---|---|---|---|---|---|---|
| All: 0.056 - 18 μm | Phthalate | 0 - 67.02 | 8.68 (13.77) | 1.97 - 67.02 | 16.75 (24.80) | 17.36 - 45.30 | 26.94 (15.91) | 0 - 14.72 | 4.79 (4.37) | 0-25.03 | 5.72 (7.41) |
| | Adipate | 0 - 43.83 | 7.60 (9.38) | 0 - 20.18 | 9.08 (8.82) | 0.24 - 19.56 | 8.47 (9.97) | 0 - 13.00 | 4.22 (3.77) | 0 - 43.83 | 13.18 (14.66) |
| | Succinate | 0 - 116.28 | 9.53 (22.25) | 0 - 116.28 | 21.61 (43.10) | 0 - 14.31 | 7.57 (7.19) | 0 - 62.83 | 7.14 (13.63) | 0 - 20.14 | 4.73 (7.43) |
| | Maleate | 0 - 119.19 | 9.52 (19.66) | 2.56 - 58.39 | 18.68 (14.89) | 0.19 - 8.45 | 3.81 (4.23) | 0 - 14.42 | 1.65 (3.65) | 2.30 - 119.19 | 19.44 (34.04) |
| | Oxalate | 37.67 - 472.82 | 148.59 (94.26) | 49.83 - 472.82 | 177.86 (139.41) | 179.42 - 365.10 | 252.17 (99.15) | 51.62 - 421.82 | 143.65 (75.76) | 37.67 - 214.62 | 110.21 (62.06) |
| | MSA | 0.10 - 23.33 | 5.40 (5.23) | 2.77 - 23.33 | 10.01 (6.60) | 0.16 - 16.14 | 5.55 (9.18) | 0.10 - 7.45 | 3.08 (2.02) | 0.84 - 17.52 | 6.30 (5.38) |
| 0.056 - 1 μm | Phthalate | 0 - 64.53 | 5.61 (12.90) | 0.51 - 64.53 | 14.78 (24.24) | 9.14 - 39.62 | 20.23 (16.85) | 0 - 9.38 | 1.64 (2.47) | 0 - 8.51 | 2.72 (3.11) |
| | Adipate | 0 - 31.57 | 4.27 (5.84) | 0 - 15.94 | 6.10 (6.25) | 0 - 10.99 | 5.53 (5.50) | 0 - 10.64 | 2.51 (3.15) | 0 - 31.57 | 6.09 (8.80) |
| | Succinate | 0 - 108.47 | 7.35 (19.58) | 0 - 108.47 | 18.54 (38.92) | 0 - 13.54 | 7.31 (6.84) | 0 - 52.42 | 4.25 (10.36) | 0 - 15.68 | 4.32 (6.61) |
| | Maleate | 0 - 108.65 | 9.17 (18.41) | 2.56 - 57.73 | 18.39 (14.92) | 0.19 - 8.45 | 3.81 (4.23) | 0 - 14.42 | 1.63 (3.64) | 2.30 - 108.65 | 18.27 (31.26) |
| | Oxalate | 16.21 - 318.49 | 93.30 (61.81) | 29.96 - 256.72 | 108.26 (75.45) | 96.84 - 250.78 | 165.57 (78.28) | 26.11 - 318.49 | 90.60 (58.45) | 16.21 - 151.79 | 69.58 (39.62) |
| | MSA | 0 - 21.32 | 5.01 (4.93) | 2.41 - 21.32 | 9.25 (6.15) | 0.08 - 15.58 | 5.33 (8.88) | 0 - 7.45 | 2.90 (2.09) | 0.84 - 16.22 | 5.72 (5.09) |
| 1 - 18 μm | Phthalate | 0 - 16.52 | 3.07 (3.27) | 0 - 4.07 | 1.98 (1.65) | 5.43 - 9.03 | 6.71 (2.01) | 0 - 9.42 | 3.15 (2.63) | 0 - 16.52 | 3.00 (4.99) |
| | Adipate | 0 - 26.00 | 3.34 (4.94) | 0 - 7.87 | 2.98 (3.17) | 0 - 8.56 | 2.93 (4.87) | 0 - 8.07 | 1.71 (2.20) | 0 - 26.00 | 7.10 (7.98) |
| | Succinate | 0 - 21.18 | 2.18 (4.53) | 0 - 16.02 | 3.06 (4.90) | 0 - 0.77 | 0.26 (0.44) | 0 - 21.18 | 2.89 (5.36) | 0 - 5.33 | 0.41 (1.48) |
| | Maleate | 0 - 10.54 | 0.35 (1.51) | 0 - 2.30 | 0.29 (0.70) | 0 | 0 | 0 - 0.45 | 0.02 (0.09) | 0 - 10.54 | 1.17 (2.94) |
| | Oxalate | 6.27 - 216.10 | 55.29 (38.52) | 19.87 - 216.10 | 69.60 (67.47) | 62.90 - 114.32 | 86.60 (25.95) | 18.51 - 104.88 | 53.05 (22.89) | 6.27 - 103.58 | 40.63 (28.89) |
| | MSA | 0 - 2.00 | 0.40 (0.50) | 0 - 2.00 | 0.75 (0.55) | 0 - 0.56 | 0.21 (0.30) | 0 - 1.58 | 0.18 (0.36) | 0 - 1.93 | 0.58 (0.56) |




**Table 2:** Contributions of the five positive matrix factorization (PMF) source factors to each individual organic acid and MSA.

|  | Combustion | Biomass Burning | Crustal | Sea Salt | Waste Processing |
|---|---|---|---|---|---|
| **Phthalate** | 27.4 % | 49.5 % | 13.3 % | 9.9 % | 0 % |
| **Adipate** | 32.9 % | 26.4 % | 35.9 % | 4.7 % | 0 % |
| **Succinate** | 0 % | 90.3 % | 9.7 % | 0 % | 0 % |
| **Maleate** | 69.7 % | 0 % | 0.2 % | 0 % | 30.1 % |
| **Oxalate** | 32.9 % | 25.4 % | 31.2 % | 0 % | 10.5 % |
| **MSA** | 57.4 % | 41.2 % | 0.1 % | 0 % | 1.4 % |





**Table 3:** Pearson's correlation matrices (r values) of water-soluble species for submicrometer
(0.056 – 1.0 µm) and supermicrometer (1.0 – 18 µm) sizes. Blank boxes indicate p-values
exceeding 0.05 and thus deemed to be statistically insignificant. Ad – adipate, Su – succinate,
Ma – maleate, Ox – oxalate, Ph – phthalate. A similar correlation matrix for the full size range
(0.056 – 18 µm) is in Table S6.

**< 1 µm**

|      | Al   | Ti   | K    | Rb   | V    | Ni   | As   | Cd   | Pb   | Na   | Cl   | NO3  | SO4  | MSA  | Ad   | Su   | Ma   | Ox   | Ph   |
|------|------|------|------|------|------|------|------|------|------|------|------|------|------|------|------|------|------|------|------|
| Al   | 1.00 |      |      |      |      |      |      |      |      |      |      |      |      |      |      |      |      |      |      |
| Ti   |      | 1.00 |      |      |      |      |      |      |      |      |      |      |      |      |      |      |      |      |      |
| K    | 0.91 |      | 1.00 |      |      |      |      |      |      |      |      |      |      |      |      |      |      |      |      |
| Rb   | 0.44 |      | 0.48 | 1.00 |      |      |      |      |      |      |      |      |      |      |      |      |      |      |      |
| V    |      | 0.28 |      | 0.36 | 1.00 |      |      |      |      |      |      |      |      |      |      |      |      |      |      |
| Ni   |      | 0.47 |      | 0.40 | 0.89 | 1.00 |      |      |      |      |      |      |      |      |      |      |      |      |      |
| As   |      |      |      |      |      |      | 1.00 |      |      |      |      |      |      |      |      |      |      |      |      |
| Cd   |      |      |      |      | 0.64 | 0.68 |      | 1.00 |      |      |      |      |      |      |      |      |      |      |      |
| Pb   | 0.41 |      | 0.32 | 0.27 | 0.28 | 0.40 |      | 0.42 | 1.00 |      |      |      |      |      |      |      |      |      |      |
| Na   |      |      |      |      |      |      |      |      |      | 1.00 |      |      |      |      |      |      |      |      |      |
| Cl   | 0.90 |      | 0.99 | 0.39 |      |      |      |      | 0.30 |      | 1.00 |      |      |      |      |      |      |      |      |
| NO3  | 0.76 |      | 0.82 | 0.28 |      |      |      |      | 0.84 |      | 1.00 |      |      |      |      |      |      |      |      |
| SO4  |      |      |      | 0.42 | 0.48 | 0.40 |      |      |      |      |      |      | 1.00 |      |      |      |      |      |      |
| MSA  |      |      |      | 0.39 |      |      |      |      |      |      | 0.60 |      |      | 1.00 |      |      |      |      |      |
| Ad   |      |      |      |      |      |      |      |      |      |      |      |      |      |      | 1.00 |      |      |      |      |
| Su   |      | 0.31 |      | 0.67 |      |      |      |      |      |      |      |      | 0.45 | 0.67 | 0.33 | 1.00 |      |      |      |
| Ma   |      |      |      |      |      |      |      |      |      |      |      |      |      |      | 0.32 |      | 1.00 |      |      |
| Ox   |      | 0.35 |      | 0.70 | 0.47 | 0.53 |      |      |      |      |      |      | 0.72 | 0.47 |      | 0.69 |      | 1.00 |      |
| Ph   |      | 0.37 |      | 0.53 |      |      |      |      |      |      |      |      | 0.39 | 0.67 | 0.45 | 0.82 |      | 0.57 | 1.00 |

**> 1 µm**

|      | Al   | Ti   | K    | Rb   | V    | Ni   | As   | Cd   | Pb   | Na   | Cl   | NO3  | SO4  | MSA  | Ad   | Su   | Ma   | Ox   | Ph   |
|------|------|------|------|------|------|------|------|------|------|------|------|------|------|------|------|------|------|------|------|
| Al   | 1.00 |      |      |      |      |      |      |      |      |      |      |      |      |      |      |      |      |      |      |
| Ti   | 0.56 | 1.00 |      |      |      |      |      |      |      |      |      |      |      |      |      |      |      |      |      |
| K    |      |      | 1.00 |      |      |      |      |      |      |      |      |      |      |      |      |      |      |      |      |
| Rb   | 0.62 |      | 0.48 | 1.00 |      |      |      |      |      |      |      |      |      |      |      |      |      |      |      |
| V    |      | 0.40 |      | 0.31 | 1.00 |      |      |      |      |      |      |      |      |      |      |      |      |      |      |
| Ni   |      | 0.30 |      |      |      | 1.00 |      |      |      |      |      |      |      |      |      |      |      |      |      |
| As   |      | 0.37 |      |      | 0.33 |      | 1.00 |      |      |      |      |      |      |      |      |      |      |      |      |
| Cd   |      |      |      |      | 0.66 | 0.41 | 0.34 | 1.00 |      |      |      |      |      |      |      |      |      |      |      |
| Pb   | 0.43 | 0.45 |      | 0.36 | 0.51 | 0.45 |      | 0.65 | 1.00 |      |      |      |      |      |      |      |      |      |      |
| Na   | 0.49 | 0.42 |      |      |      |      |      |      |      | 1.00 |      |      |      |      |      |      |      |      |      |
| Cl   | 0.45 | 0.48 |      |      |      |      |      |      |      | 0.90 | 1.00 |      |      |      |      |      |      |      |      |
| NO3  | 0.38 |      |      | 0.32 | 0.41 |      |      |      |      | 0.64 | 0.30 | 1.00 |      |      |      |      |      |      |      |
| SO4  | 0.39 |      | 0.81 | 0.64 |      |      |      |      |      | 0.37 | 0.29 | 0.36 | 1.00 |      |      |      |      |      |      |
| MSA  |      |      |      |      |      |      |      |      |      | 0.32 |      |      |      | 1.00 |      |      |      |      |      |
| Ad   |      |      |      |      |      |      |      |      |      |      |      |      |      |      | 1.00 |      |      |      |      |
| Su   | 0.39 |      |      | 0.28 |      |      |      |      |      | 0.30 |      |      |      |      |      | 1.00 |      |      |      |
| Ma   |      |      |      |      |      |      |      |      |      |      |      |      |      |      | 0.57 |      | 1.00 |      |      |
| Ox   | 0.59 | 0.29 |      | 0.48 |      |      |      |      |      | 0.45 |      | 0.59 | 0.35 |      |      | 0.45 |      | 1.00 |      |
| Ph   |      | 0.29 |      |      |      |      |      |      |      |      | 0.34 |      |      |      | 0.30 |      |      |      | 1.00 |


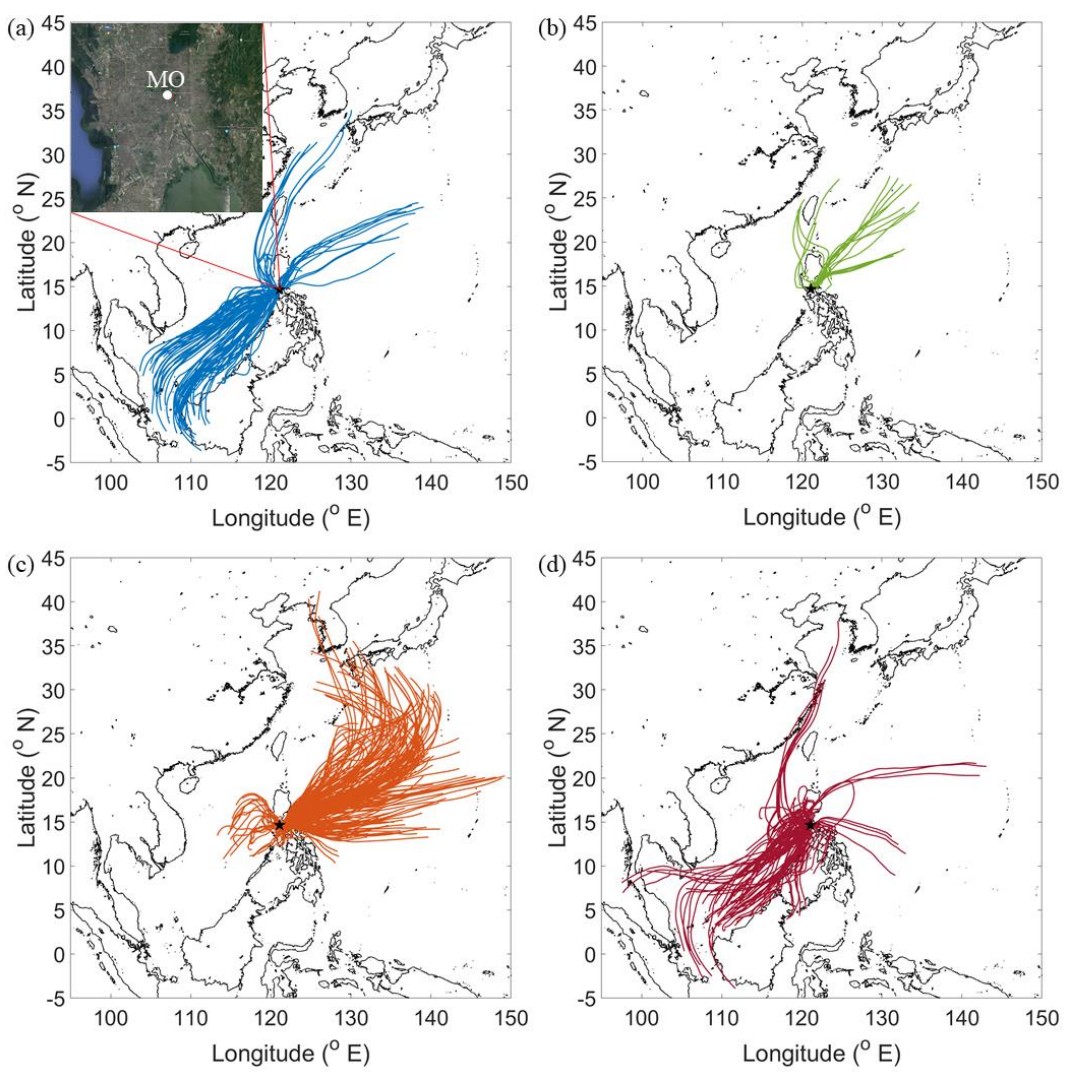


**Figure 1:** HYSPLIT back-trajectories for four seasons: (a) 2018 southwest monsoon (SWM18), (b) transitional period (Trans), (c) northeast monsoon (NEM), and (d) 2019 southwest monsoon (SWM19). Results shown are based on 72-hour back-trajectories collected every 6 h during sampling periods. The top left corner of panel (a) zooms in on Metro Manila with Manila Observatory (MO) marked. The black star in each panel represents the sampling site. Map data: © Google Earth, Maxar Technologies, CNES/Airbus, Data SIO, NOAA, U.S. Navy, NGA, GEBCO.





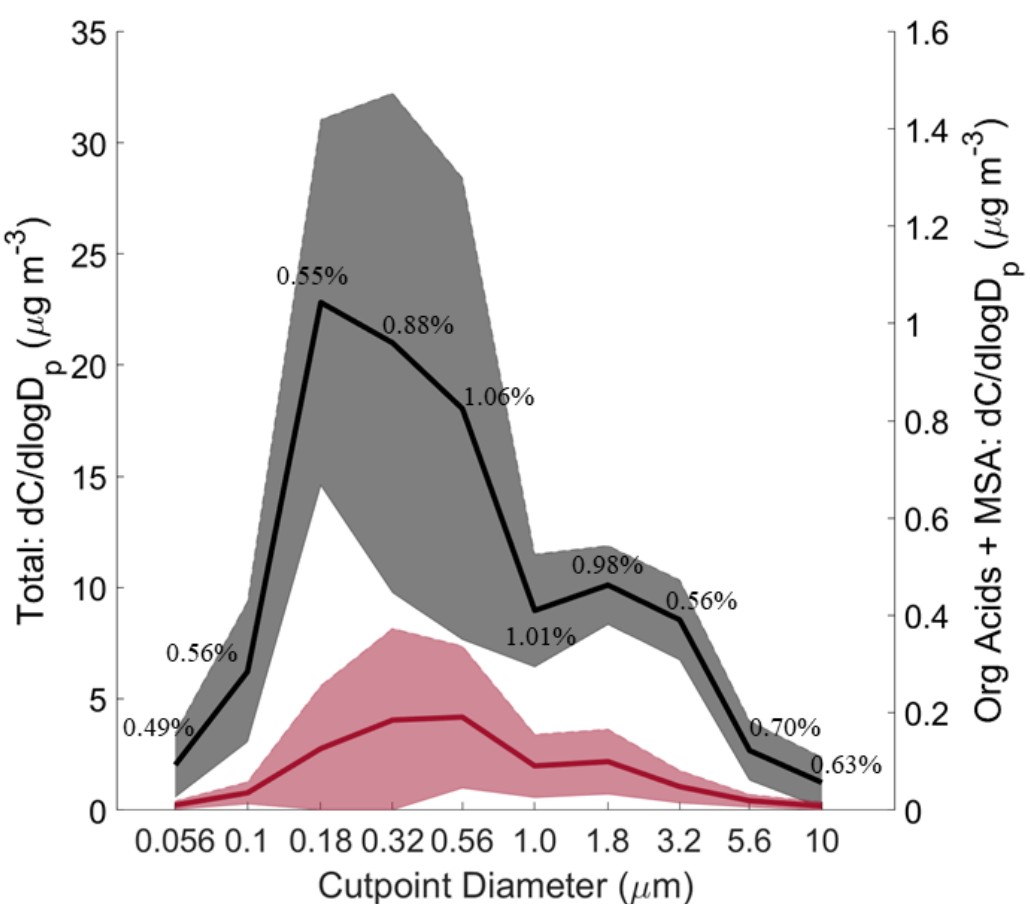

**Figure 2:** Size-resolved comparison of total mass versus the sum of measured organic acids and MSA. The black curve represents total mass and the red curve represents the summed organic acids and MSA. Solid lines are the averages and shaded areas are one standard deviation. These plots were made based on data from the 11 MOUDI chemical sets with accompanying gravimetric measurements. The average percent contribution of the organic acids and MSA to total mass is provided for each size bin. Refer to Fig. S1 for the seasonally-resolved version of this figure.



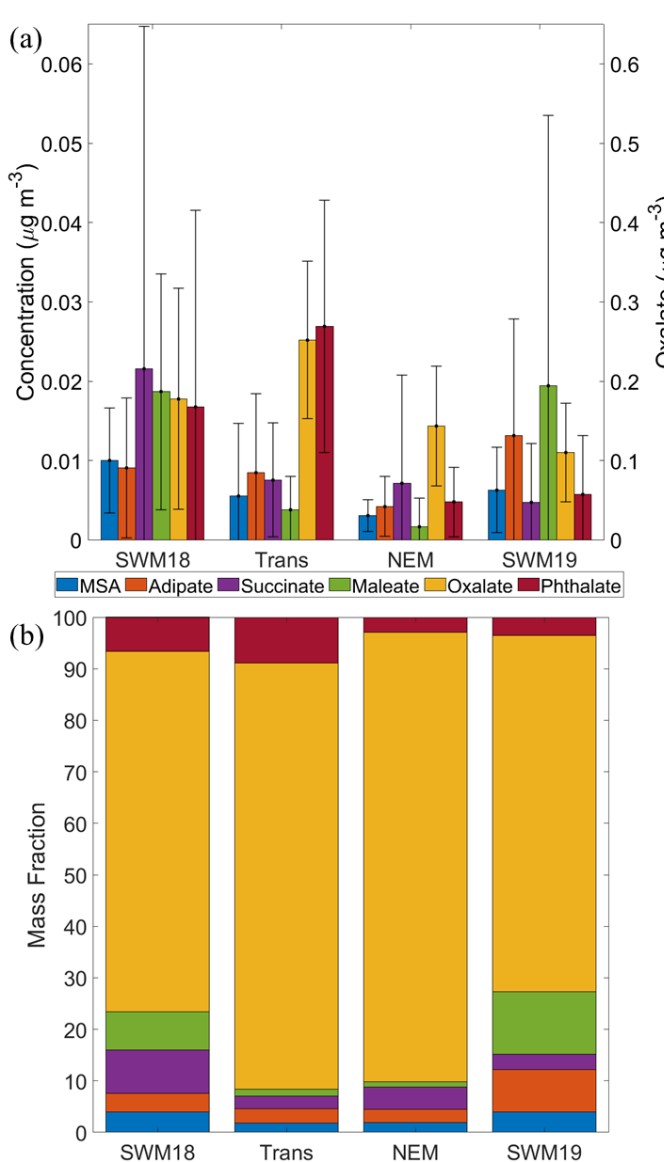

**Figure 3:** (a) Average concentrations (0.056 – 18 μm) for (left y-axis) MSA, adipate, succinate, maleate, and phthalate, in addition to (right y-axis) oxalate. Black bars represent one standard deviation. (b) Percentage relative mass abundance of organic acids and MSA separated based on season.

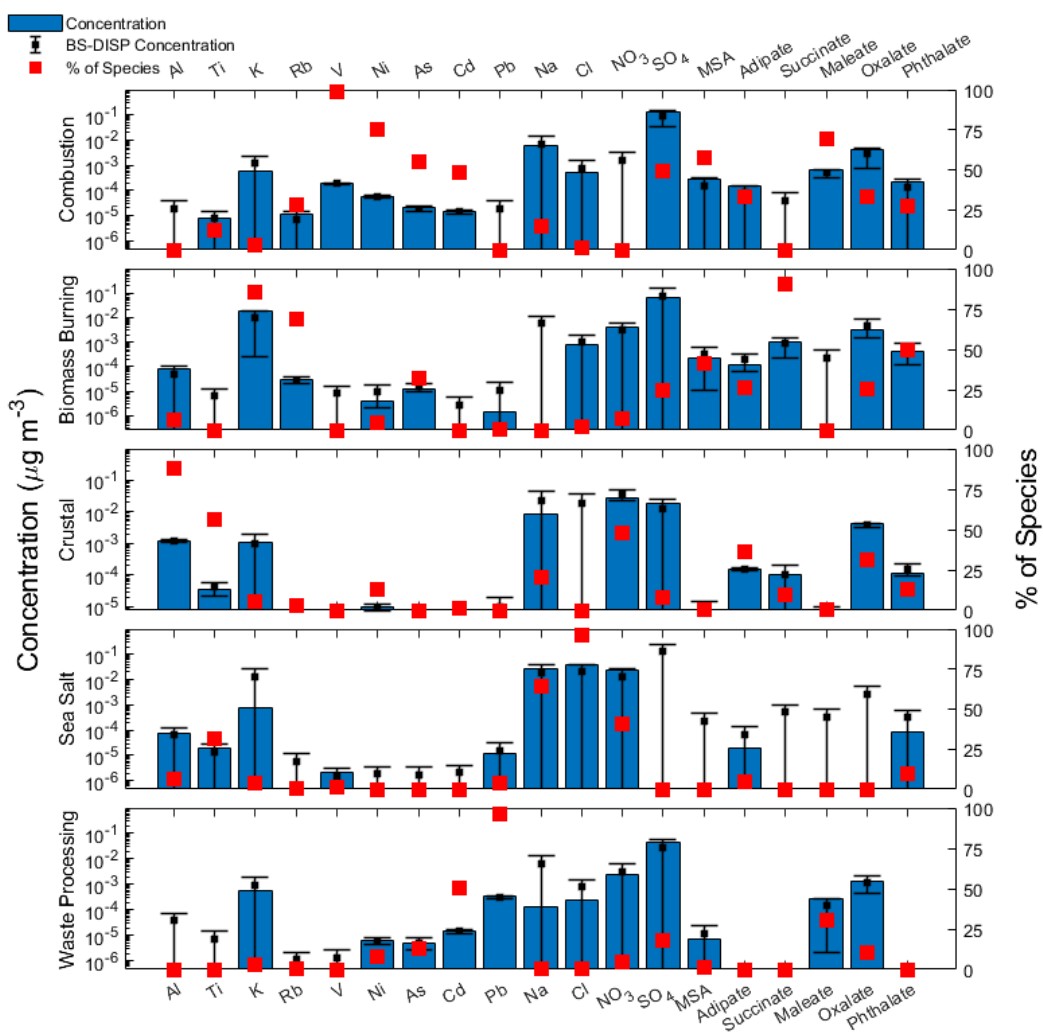

1538

**Figure 4:** Source factor profiles from positive matrix factorization (PMF) analysis. Blue bars
represent the mass concentration contributed to the respective factor, red filled squares represent
the percentage of total species associated with that source factor, and black squares with error
bars represent the average, 5th, and 95th percentiles of bootstrapping with displacement (BS-
DISP) values.






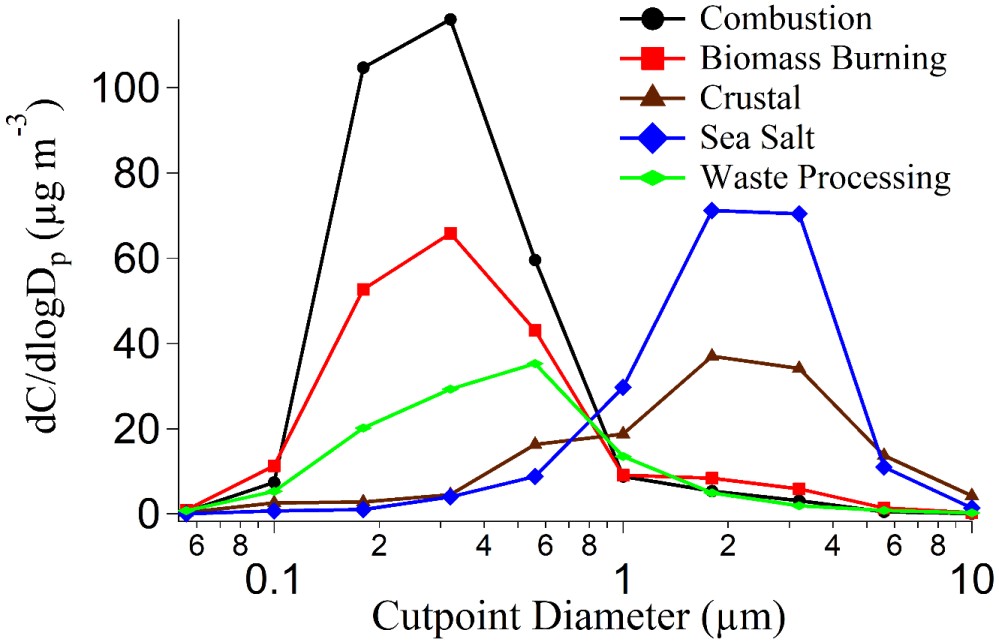

**Figure 5:** Reconstructed mass size distributions of positive matrix factorization (PMF) factors.

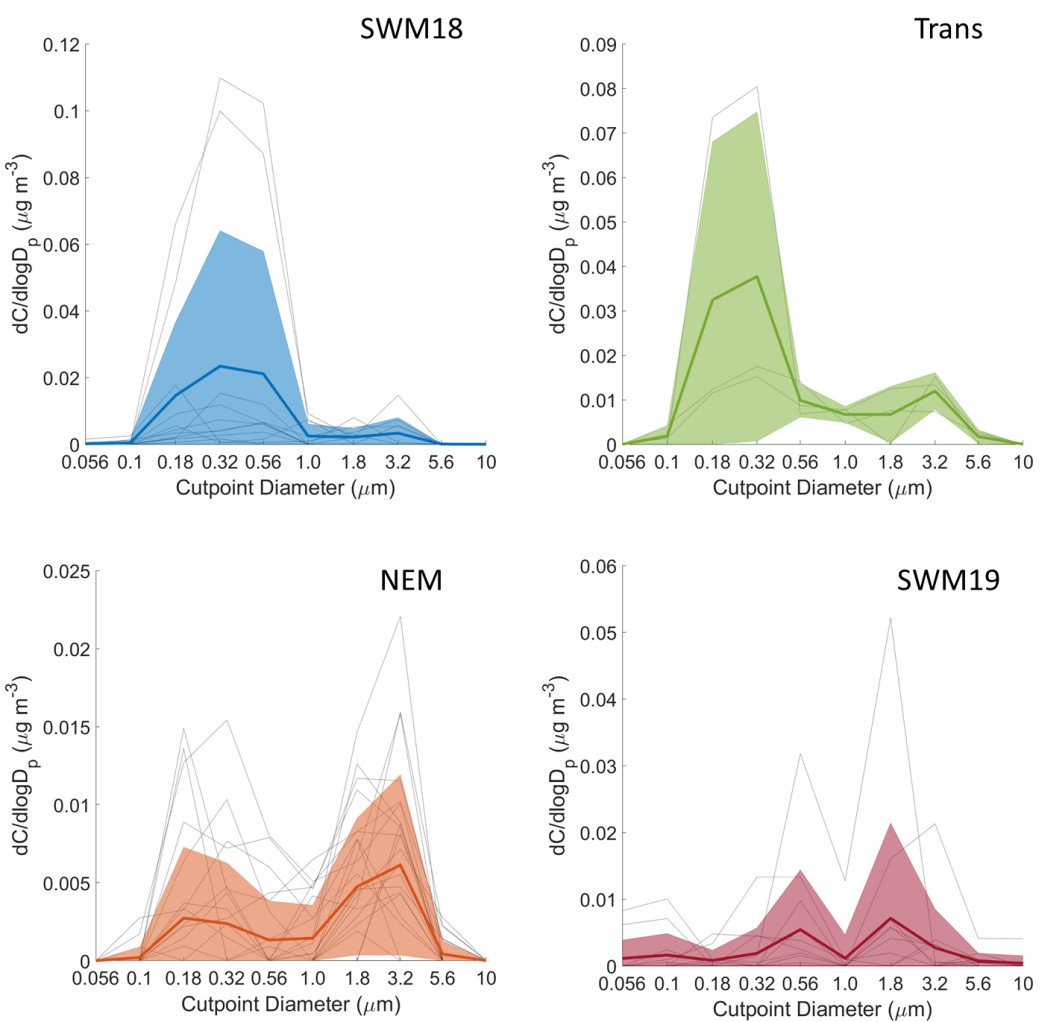


**Figure 6:** Seasonal size distributions of phthalate. Gray lines represent individual sets, dark
colored lines are the average of all seasonal distributions, and transparent colored areas represent
one standard deviation. Note that the range of concentrations presented on the y-axis for each
season varies.







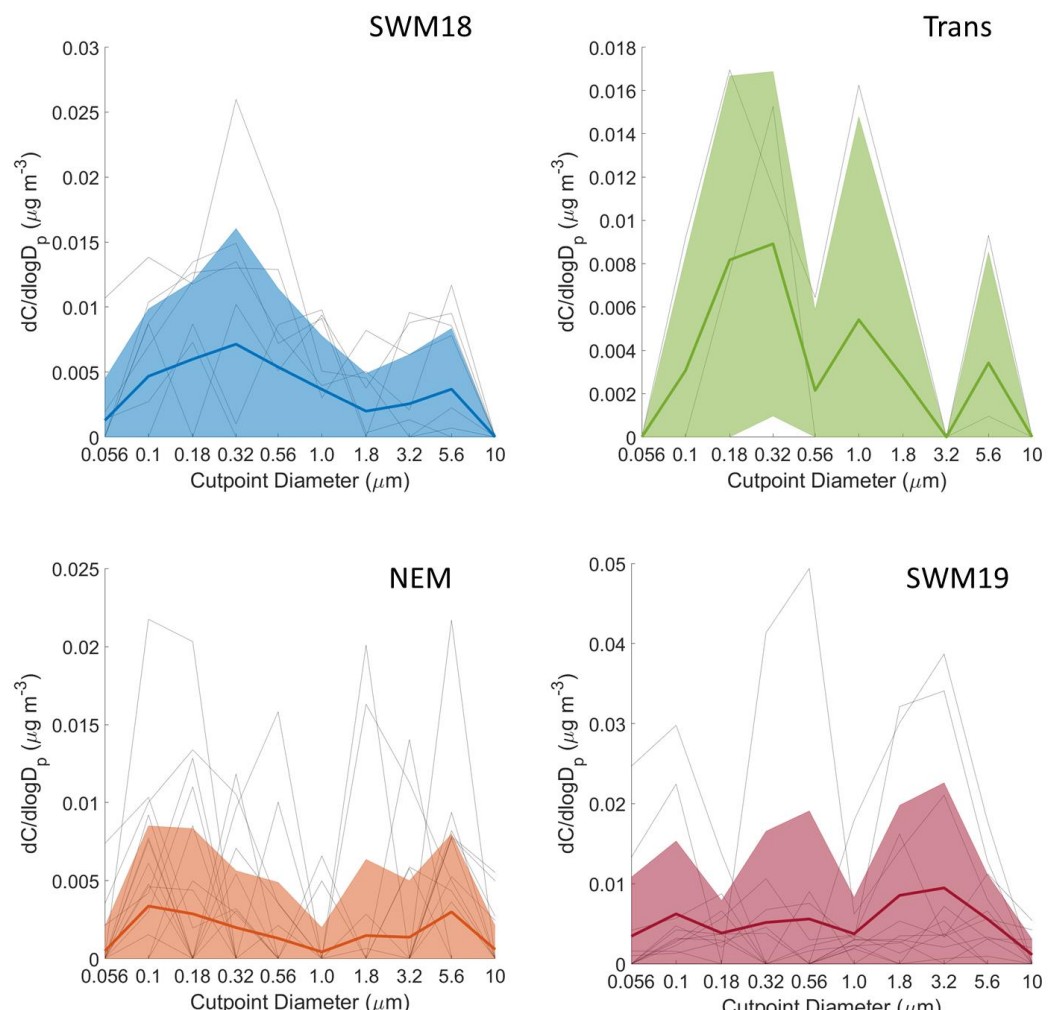


**Figure 7:** Same as Fig. 6 but for adipate.




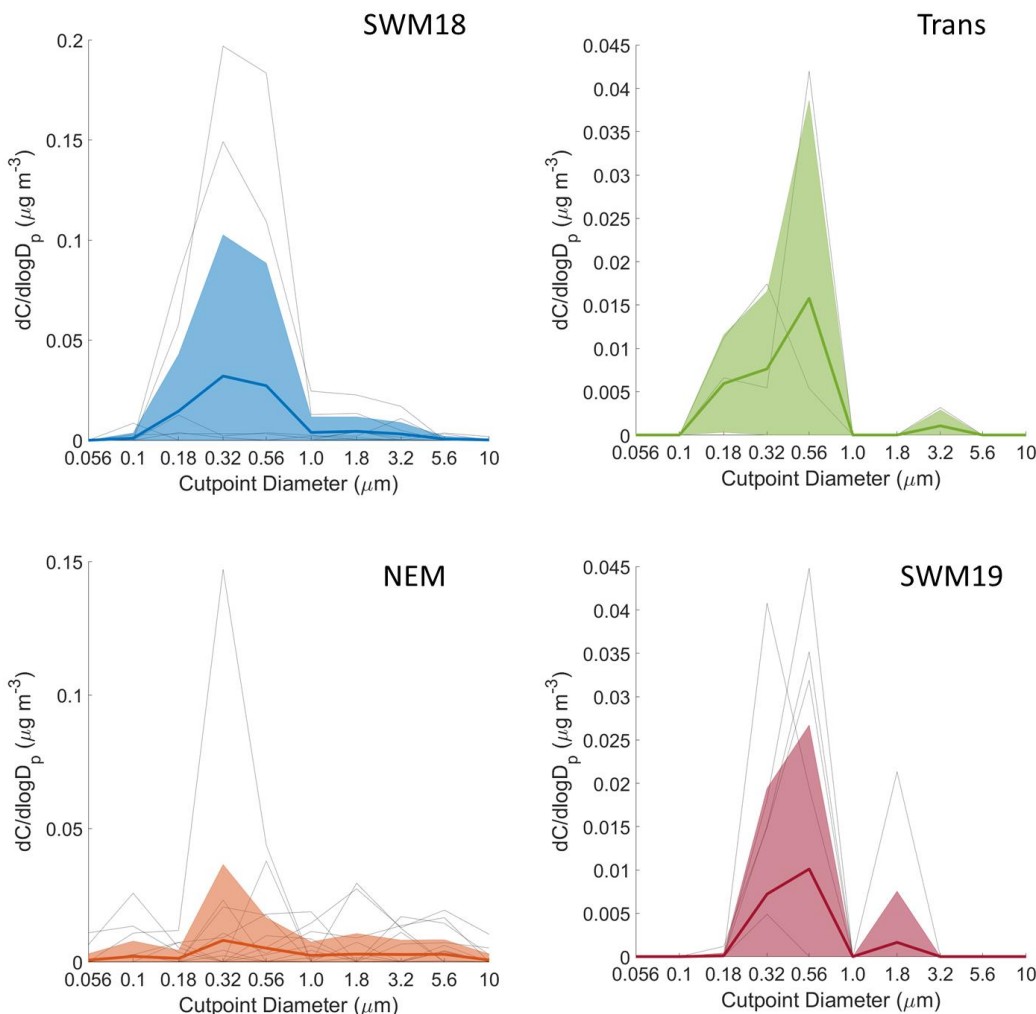


**Figure 8:** Same as Fig. 6 but for succinate.





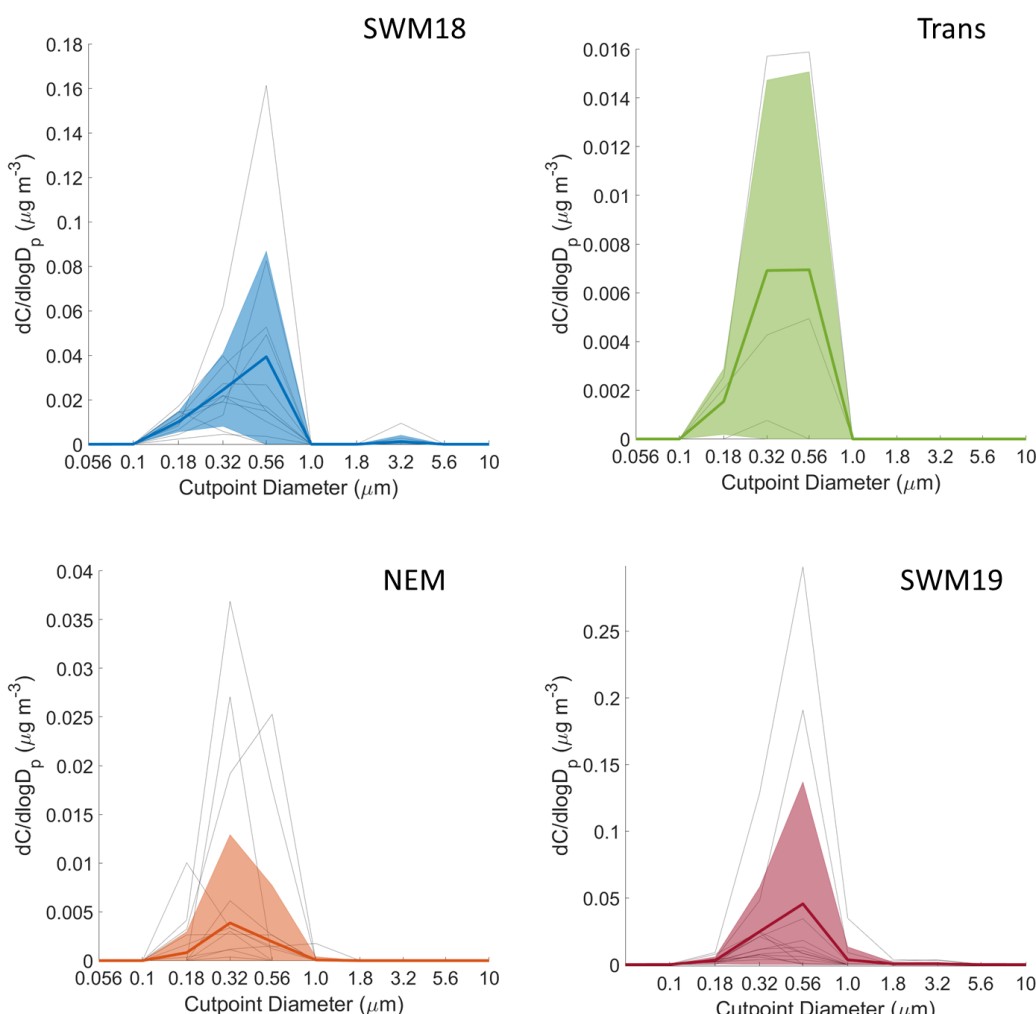


**Figure 9:** Same as Fig. 6 but for maleate.



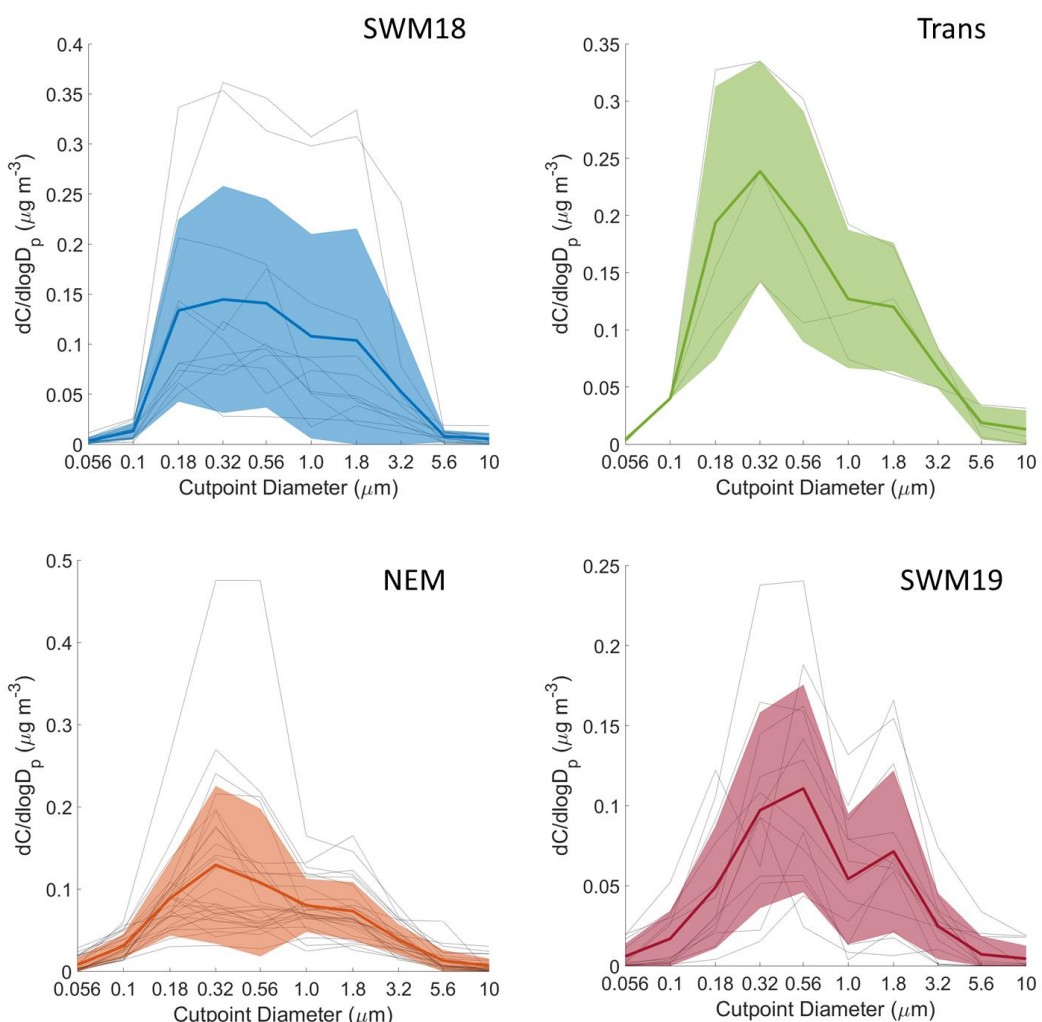


**Figure 10:** Same as Fig. 6 but for oxalate.



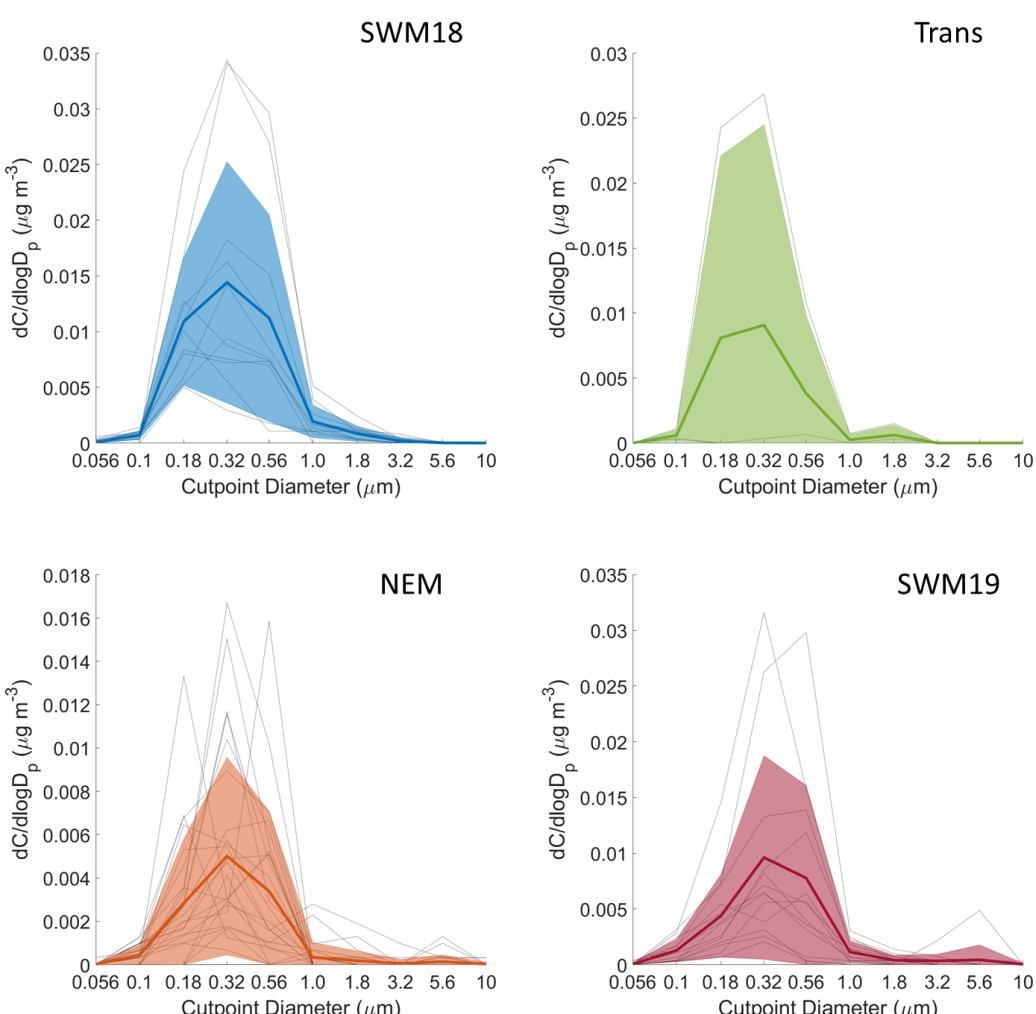


**Figure 11:** Same as Fig. 6 but for MSA.


**Figure 12:** WCWT maps of (a-e) individual organic acids and (f) MSA over the entire sampling period. These results are based on all MOUDI sizes (0.056 – 18 μm). Maps showing the seasonal results for each organic acid and MSA are shown in the Supplement (Figs. S3 – S8).