# Peer review of "Sources and characteristics of size-resolved particulate organic acids and methanesulfonate in a"

_Atmospheric Chemistry and Physics, 2020_

## Referee Comment (RC1) · Anonymous Referee #1 · 7 Sep 2020

General Comments: The paper presents aerosol composition data results in Manila during an 18-month sampling period including 2 monsoon seasons. Overall, the results provide a detailed connection between aerosol composition, pollution source and source region. Many related analyses are performed and provide for a complete picture of the measured organic acids and MSA. The authors did a very good job of weaving together such detailed analysis into one manuscript. However, minor revisions are needed to help with the readability of the paper.

Specific Comments:

1) The fact that the PMF did not include sea salt as a contributor for MSA is alarming

and needs to be addressed more (first described in lines 401-405). While it is possible that there are other sources as seen in the Beijing and California studies, the fact that there is absolutely no link between sea salt and MSA needs further analysis. Could it be that local sources of sea salt were not aged enough to allow for MSA formation (the sea salt air masses were too fresh)? What was the height of the roof where the sampler was placed? Does this possibly suggest an error in the PMF?

2) In the paper, the individual sources are linked to a source and to a season but I don't think there is a clear link between which sources dominate on whole for each season. ie: during the monsoon, biomass is strongest with a source region from the southwest... (made up results)

3) Section 2.2: include information on filter extraction

4) Line 230-231: include a reference for PMF

5) Line 253-283: this should not be shown in results. Move to new section in methods section – such as new section 2.3

6) Line 291-293: are these accumulated rain over the entire period or rain per day. If per day this should be noted. If accumulated rain, how is there variability?

7) Discussion: Many related analyses are performed and are summarized in the first paragraph of each subsection. This should be a numbered, indented list so they stand out more allowing for comparison between the different species. Currently this good discussion is presented as a long run-on sentence at the beginning of each sub-section.

8) Discussion: a long list of measurements in other locations are included for each compound (ie: Line 4750-479; 519-524; 556-561...). This is not needed as this is not a review. I would suggest including 1 or 2 relevant measurements for comparison and including the rest in the supplemental information if needed.

9) Line 630 and following: this is not a clear bi-modal distribution. Rather 2-3 modes

that run together possibly showing a similar secondary source for all sizes

10) Throughout: the use of "Trans" as an abbreviation is unneeded and a little distracting. Replace with "Transition"

11) Table 3: I think switch Table 3 and Table S6 so only the correlation for all size distributions is in the main text.

12) Figure 1: include clearer labels for each season above each panel

Technical Corrections:

1) Line 32 and throughout: 148.59 +/- 94.26 has too many significant figures. Your variability should be trimmed to 1 or 2 significant figures and then your average adjusted to match the decimal place. That is it should be 149 +/- 94. Other places noted (not a complete list): lines 289-290; line 337-338

2) Line 33: define PMF

3) Line 35 & 38: line 35 remove "(SWM18)" and line 38 replace "SWM" with "monsoon"

4) Line 37: "linked to burning" should be "linked to biomass burning"

5) Line 44-48: run-on sentence. Split into two easier to understand sentences

6) Line 67: new paragraph

7) Line 132: 12.9 million

8) Line 136: word choice for quintessential – maybe use "fitting"

9) Line 181: "a subset of species were listed here, which were used for analyses." should be replaced with "a subset of species used for analyses are listed here."

10) Line 199: Replace "the roof of MO. Measured" with "the roof. Measured"

11) Line 227: should use CWT and not WCWT

12) Line 320: replace "(0.80 ± 0.66 %), ranging from 0.23 – 1.49 % across" with "(0.80 ± 0.66 %) across" – this was redundant with later in the paragraph

13) Line 364-380: suggest making these a numbered list so they stand out more

---

## Referee Comment (RC2) · Anonymous Referee #2 · 2 Oct 2020

This manuscript reports the analytical results of five organic acids and MSA in the size-resolved aerosols from Manila and discusses their sources and characteristics based on the correlation analyses with a 16-month dataset of inorganic ions and trace metals. Because of no studies on organic acids in the aerosols from Manila areas, authors' dataset may be of interest for the community of atmospheric chemistry. However, this manuscript is very descriptive and sometimes redundant. As authors pointed, the concentration levels of organic acids are surprisingly low compared with other polluted areas (lines 22-23, 704-706) although authors did not provide a reasonable explanation. My major concern is that the concentrations of organic acids and MSA could be seriously underestimated due to biodegradation because there is no description in

the text to avoid possible decomposition of organic acids and other chemical species during sample storage and analytical procedure. Another concern is that, although authors mentioned "Little is reported in terms of the size-resolved nature of organic acids and MSA .." (line 92-93), there are several studies that reported size distributions of organic acids (see below for details). The above points should be clarified in the revision before the consideration for the decision on a possible acceptance to ACP. More specific comments are followed:

1. Lines 31-33. Oxalate was approximately an order of magnitude more abundant... Which size fraction are you talking about? Total (<0.056 to >18 $\mu$m)?

2. Line 51 and others. Some of the references are not properly cited. For example, in line 51, authors cited Kondo et al. (2011) in the discussion of organic acids. However, Kondo et al. (2011) focused on black carbon but not for organic acids. Please check the possible mistakes in referring previous citations.

3. Lines 92-93. There are several studies on the size-segregated dicarboxylic acids from different regions in the world. For example:

Mochida, M., N. Umemoto, K. Kawamura, H. Lim, and B. J. Turpin (2007), Bimodal size distributions of various organic acids and fatty acids in the marine atmosphere: Influence of anthropogenic aerosols, Asian dusts, and sea spray off the coast of East Asia, J. Geophys. Res., 112, D15209, doi:10.1029/2006JD007773.

Kawamura K., M. Narukawa, S.-M. Li and L. A. Barrie, Size distributions of dicarboxylic acids and inorganic ions in atmospheric aerosols collected during polar sunrise in the Canadian High Arctic. J. Geophys. Res., 112, D10307, doi:10.1029/2006JD008244, 2007.

Gehui Wang, Kimitaka Kawamura, Mingjie Xie, Shuyuan Hu, Junji Cao, Zhisheng An, John G. Waston and Judith C. Chow, Organic Molecular Compositions and Size Distributions of Chinese Summer and Autumn Aerosols from Nanjing: Characteristic Haze

Event Caused by Wheat Straw Burning, Environ. Sci. Technol., 43 (17), 6493–6499, 2009.

Gehui Wang, Kimitaka Kawamura, Mingjie Xie, Shuyuan Hu, and Zifa Wang, Water-soluble organic compounds in PM2.5 and size-segregated aerosols over Mt. Tai in North China Plain, J. Geophys. Res., 114, D19208, doi:10.1029/2008JD011390, 2009.

Smita Agarwal, Shankar Gopala Aggarwal, Kazuhiro Okuzawa, and Kimitaka Kawamura, Size Distributions of Dicarboxylic Acids, Ketoacids, $\alpha$-Dicarbonyls, Sugars, WSOC, OC, EC and Inorganic Ions in Atmospheric Particles Over Northern Japan: Implication for Long-Range Transport of Siberian Biomass Burning and East Asian Polluted Aerosols, Atmos. Chem. Phys., 10, 5839-5858, 2010.

Gehui Wang, Kimitaka Kawamura, Mingjie Xie, Shuyuan Hu, Jianjun Li, Bianhong Zhou, Junji Cao, Zhisheng An, Selected water-soluble organic compounds found in sized-resolved aerosols collected from urban, mountain, and marine atmospheres over East Asia, Tellus, 63B, 371-381, 2011.

Miyazaki, Y., K. Kawamura, and M. Sawano (2010), Size distributions and chemical characterization of water‐soluble organic aerosols over the western North Pacific in summer, J. Geophys. Res., 115, D23210, doi:10.1029/2010JD014439.

Dhananjay Kumar Deshmukh, Kimitaka Kawamura and Manas Kanti Deb, Dicarboxylic acids, $\omega$-oxocarboxylic acids, ïĄ̧-dicarbonyls, WSOC, OC, EC and inorganic ions in wintertime size-segregated aerosols from central India: Sources and formation processes, Chemosphere, 161, 27-42, 2016.

Those references should be cited.

4. Method section. There is no description on the sample storage from sampling to chemical analysis. Did you store the samples at room temperature or at -20°C in a freezer? After the water extraction of the sample, how long did you store the water extracts at room temperature? Storage of filter samples and water extracts at

room temperature may be subject to biodegradation of organic acids. In Manila, high humidity conditions may provide more moistures to the aerosols. Please provide the information of storage of samples and extracts.

5. Results section. Some paragraphs are too long and redundant (e.g., 31 lines in the paragraph starting at line 253). By reorganizing the paragraphs and rephrasing the sentences, authors could improve them to become more readable.

6. Lines 340-344. Authors mentioned lower concentrations of oxalate and MSA than expected for a megacity Manila. This point should be critically verified including the discussion of methods used; potential biodegradation of organic acids and MSA during sample storage and analytical protocol.

7. Discussion. What is the reason to start the discussion with phthalate that is not the major organic acid? Why do you discuss oxalate at the end that is the most abundant organic acid? I wondered if you could improve the discussion section by reorganizing the order of the compounds.

8. Lines 514-515. How do you explain the crustal sources (35.9%) for adipate? In other places in the text, authors discuss the combustion sources of adipic acid (lines 261-262).

9. Lines 621-622 and 704-706. Again how do you explain the "surprisingly low concentrations of oxalate" ? This reviewer is skeptical about data quality; i.e., a potential loss of organic acids and MSA due to biodegradation during the sample storage and analytical procedure used in this study.

10. Lines 695-698. Is it possible to discuss the fraction of organic acids and MSA on a carbon basis? Did you measure TOC or WSOC?

---

## Author Comment (AC1) · 12 Oct 2020

Response: We thank the reviewers for thoughtful suggestions and constructive criticism that have helped us improve our manuscript. Below we provide responses to reviewer concerns and suggestions in blue font. All changes to the manuscript can be identified in the version submitted using Track Changes.

**Anonymous Referee #1:**

General Comments: The paper presents aerosol composition data results in Manila during an 18-month sampling period including 2 monsoon seasons. Overall, the results provide a detailed connection between aerosol composition, pollution source and source region. Many related analyses are performed and provide for a complete picture of the measured organic acids and MSA. The authors did a very good job of weaving together such detailed analysis into one manuscript. However, minor revisions are needed to help with the readability of the paper.

Specific Comments:
1) The fact that the PMF did not include sea salt as a contributor for MSA is alarming and needs to be addressed more (first described in lines 401-405). While it is possible that there are other sources as seen in the Beijing and California studies, the fact that there is absolutely no link between sea salt and MSA needs further analysis. Could it be that local sources of sea salt were not aged enough to allow for MSA formation (the sea salt air masses were too fresh)? What was the height of the roof where the sampler was placed? Does this possibly suggest an error in the PMF?

Response: The fact that MSA is not a contributor to sea salt can be initially alarming, however, there are at least two possible explanations for this. First of all, as you pointed to the sea salt in the region is relatively fresh and likely not very aged, thus yielding low MSA concentrations. The second explanation is that Manila is not like any other coastal site because it is a mega-center for urban and burning emissions and is also a receptor for transported plumes of aerosol types such as smoke. Consequently, concentrations of MSA from these other non-marine sources could be much higher. Because of these reasons it is understandable how the model does not assign MSA concentrations to sea salt, but to other sources. We believe our text in Section 5.6 about MSA is sufficient and requires no revision.

2) In the paper, the individual sources are linked to a source and to a season but I don't think there is a clear link between which sources dominate on whole for each season. ie: during the monsoon, biomass is strongest with a source region from the southwest. . . (made up results)

Response: Our goal was to show what organic acids are associated with what sources, not what sources dominate each season. Our current analyses does not determine what sources dominate each season. We don't believe this comment requires any revision.

3) Section 2.2: include information on filter extraction

Response: A detailed version of the extraction method can be seen in Stahl et al. (2020), however, we added a brief summary of the extraction for convenience, which reads:

Lines 166-172: "Substrates were stored in a freezer at -20 °C after samples were collected from the MOUDI until extractions could be carried out. The stored substrates were then extracted by sonication in Milli-Q water (18.2 MΩ-cm) for 30 minutes. After sonication, solutions were immediately analyzed to prevent degradation while the remaining extracts were stored in a refrigerator for additional analyses."

4) Line 230-231: include a reference for PMF
Response: Done.

Line 237: "… concentration budgets of the species discussed in this work (Paatero and Tapper, 1994)."

5) Line 253-283: this should not be shown in results. Move to new section in methods section – such as new section 2.3
Response: We created a new Section 3 between Methods and Results called "3. Background on Measured Acids".

6) Line 291-293: are these accumulated rain over the entire period or rain per day. If per day this should be noted. If accumulated rain, how is there variability?
Response: We are referring to accumulated rain over the sampling period for any given set and then averaged per season. There is variability between sets where some sets might not have any rain, while others can have substantial amounts. We don't believe this comment requires any revision.

7) Discussion: Many related analyses are performed and are summarized in the first paragraph of each subsection. This should be a numbered, indented list so they stand out more allowing for comparison between the different species. Currently this good discussion is presented as a long run-on sentence at the beginning of each subsection.
Response: Done.

8) Discussion: a long list of measurements in other locations are included for each compound (ie: Line 4750-479; 519-524; 556-561. . .). This is not needed as this is not a review. I would suggest including 1 or 2 relevant measurements for comparison and including the rest in the supplemental information if needed.
Response: The reason for a large list of measurements is to show how our measurements compare to other regions around the world as there is variability for different species. We feel this helps readers gain an appreciation for how Manila compares to other sites.

9) Line 630 and following: this is not a clear bi-modal distribution. Rather 2-3 modes that run together possibly showing a similar secondary source for all sizes
Response: It is hard to determine for sure whether or not there are 3 modes where 2 modes are overlapping. We can tell for sure there are at least 2 notable modes, which is why we say "appeared". But we still added this line to address this concern:

Lines 656-657: "Note that the modes discussed here represent the most pronounced ones but others could have been present too reflecting other sources."

10) Throughout: the use of "Trans" as an abbreviation is unneeded and a little distracting. Replace with "Transition"
Response: Done.

11) Table 3: I think switch Table 3 and Table S6 so only the correlation for all size distributions is in the main text.
Response: We thought that it would be more impactful to show correlations of both sub- and super-micrometer ranges. Additionally, since there is a lot of mass in the submicrometer range the correlations for the entire size range and submicrometer size range are near identical. As a result we would like to push for keeping it the way it was.

12) Figure 1: include clearer labels for each season above each panel
Response: Done.

Technical Corrections:
1) Line 32 and throughout: 148.59 +/- 94.26 has too many significant figures. Your variability should be trimmed to 1 or 2 significant figures and then your average adjusted to match the decimal place. That is it should be 149 +/- 94. Other places noted (not a complete list): lines 289-290; line 337-338
Response: Done.

2) Line 33: define PMF
Response: Done. The line now reads:
Lines 33-35: "Both positive matrix factorization (PMF) and correlation analysis is conducted with tracer species to investigate the possible sources for organic acids and MSA."

3) Line 35 & 38: line 35 remove "(SWM18)" and line 38 replace "SWM" with "monsoon"
Response: Done.

4) Line 37: "linked to burning" should be "linked to biomass burning"
Response: Done, the line now reads:
Lines 37-38: "Peculiarly, MSA had negligible contributions from marine sources but instead was linked to biomass burning and combustion."

5) Line 44-48: run-on sentence. Split into two easier to understand sentences
Response: Done and now reads:
Lines 45-48: "Oxalate's strong association with sulfate in the submicrometer mode supports an aqueous-phase formation pathway for the study region. However, high concentration during periods of low rain and high solar radiation indicates photo-oxidation is an important formation pathway."

6) Line 67: new paragraph
Response: Done.

7) Line 132: 12.9 million
Response: Done.

8) Line 136: word choice for quintessential – maybe use "fitting"

Response: Done, the line now reads:
Lines 137: "Because of these reasons, Metro Manila is a fitting location for examining locally…"

9) Line 181: "a subset of species were listed here, which were used for analyses." should be replaced with "a subset of species used for analyses are listed here."
Response: Done, the line now reads:
Lines 188-189: "It should also be noted that only a subset of species used for analyses were listed here."

10) Line 199: Replace "the roof of MO. Measured" with "the roof. Measured"
Response: Done, the line now reads:
Lines 205-206: "…Davis Vantage Pro2$^{TM}$ Plus automatic weather station, which was located on the roof."

11) Line 227: should use CWT and not WCWT
Response: Done.

12) Line 320: replace "($0.80 \pm 0.66$ %), ranging from $0.23 - 1.49$ % across" with "($0.80 \pm 0.66$ %) across" – this was redundant with later in the paragraph
Response: Done, the line now reads:
Lines 329-330: "Combined, the measured organic acids and MSA accounted for only a small part of the total cumulative mass ($0.80 \pm 0.66$ %) across the 11 individual gravimetric sets."

13) Line 364-380: suggest making these a numbered list so they stand out more
Response: Done, the sources are now numerically listed.

**Anonymous Referee #2**

This manuscript reports the analytical results of five organic acids and MSA in the size-resolved aerosols from Manila and discusses their sources and characteristics based on the correlation analyses with a 16-month dataset of inorganic ions and trace metals. Because of no studies on organic acids in the aerosols from Manila areas, authors' dataset may be of interest for the community of atmospheric chemistry. However, this manuscript is very descriptive and sometimes redundant. As authors pointed, the concentration levels of organic acids are surprisingly low compared with other polluted areas (lines 22-23, 704-706) although authors did not provide a reasonable explanation. My major concern is that the concentrations of organic acids and MSA could be seriously underestimated due to biodegradation because there is no description in the text to avoid possible decomposition of organic acids and other chemical species during sample storage and analytical procedure. Another concern is that, although authors mentioned "Little is reported in terms of the size-resolved nature of organic acids and MSA .." (line 92-93), there are several studies that reported size distributions of organic acids (see below for details). The above points should be clarified in the revision before the consideration for the decision on a possible acceptance to ACP. More specific comments are followed:
1. Lines 31-33. Oxalate was approximately an order of magnitude more abundant…

Which size fraction are you talking about? Total (<0.056 to >18 µm)?

Response: This refers to the total size range (0.056-18 µm). Additional information was added to clarify the size range. It now reads:

Lines 31-33: "Oxalate was approximately an order of magnitude more abundant than the other five species ($149 \pm 94$ ng m$^{-3}$ versus others being $< 10$ ng m$^{-3}$) across the $0.056 - 18$ µm size range."

2. Line 51 and others. Some of the references are not properly cited. For example, in line 51, authors cited Kondo et al. (2011) in the discussion of organic acids. However, Kondo et al. (2011) focused on black carbon but not for organic acids. Please check the possible mistakes in referring previous citations.

Response: The Kondo et al. (2011) reference discusses black carbon as well as organic aerosols, which was determined to be relevant to our narrative. We confirm this was not a mistake.

3. Lines 92-93. There are several studies on the size-segregated dicarboxylic acids from different regions in the world. For example:

Mochida, M., N. Umemoto, K. Kawamura, H. Lim, and B. J. Turpin (2007), Bimodal size distributions of various organic acids and fatty acids in the marine atmosphere: Influence of anthropogenic aerosols, Asian dusts, and sea spray off the coast of East Asia, J. Geophys. Res., 112, D15209, doi:10.1029/2006JD007773.

Kawamura K., M. Narukawa, S.-M. Li and L. A. Barrie, Size distributions of dicarboxylic acids and inorganic ions in atmospheric aerosols collected during polar sunrise in the Canadian High Arctic. J. Geophys. Res., 112, D10307, doi:10.1029/2006JD008244, 2007.

Gehui Wang, Kimitaka Kawamura, Mingjie Xie, Shuyuan Hu, Junji Cao, Zhisheng An, John G. Waston and Judith C. Chow, Organic Molecular Compositions and Size Distributions of Chinese Summer and Autumn Aerosols from Nanjing: Characteristic Haze Event Caused by Wheat Straw Burning, Environ. Sci. Technol., 43 (17), 6493–6499, 2009.

Gehui Wang, Kimitaka Kawamura, Mingjie Xie, Shuyuan Hu, and Zifa Wang, Water-soluble organic compounds in PM2.5 and size-segregated aerosols over Mt. Tai in North China Plain, J. Geophys. Res., 114, D19208, doi:10.1029/2008JD011390, 2009.

Smita Agarwal, Shankar Gopala Aggarwal, Kazuhiro Okuzawa, and Kimitaka Kawamura, Size Distributions of Dicarboxylic Acids, Ketoacids, _-Dicarbonyls, Sugars, WSOC, OC, EC and Inorganic Ions in Atmospheric Particles Over Northern Japan: Implication for Long-Range Transport of Siberian Biomass Burning and East Asian Polluted Aerosols, Atmos. Chem. Phys., 10, 5839-5858, 2010.

Gehui Wang, Kimitaka Kawamura, Mingjie Xie, Shuyuan Hu, Jianjun Li, Bianhong Zhou, Junji Cao, Zhisheng An, Selected water-soluble organic compounds found in sized-resolved aerosols collected from urban, mountain, and marine atmospheres over East Asia, Tellus, 63B, 371-381, 2011.

Miyazaki, Y., K. Kawamura, and M. Sawano (2010), Size distributions and chemical characterization of water-soluble organic aerosols over the western North Pacific in summer, J. Geophys. Res., 115, D23210, doi:10.1029/2010JD014439.

Dhananjay Kumar Deshmukh, Kimitaka Kawamura and Manas Kanti Deb, Dicarboxylic acids, ω-oxocarboxylic acids, α-dicarbonyls, WSOC, OC, EC, and inorganic ions in wintertime size-segregated aerosols from central India: Sources and formation processes, Chemosphere, 161, 27 42, 2016.

Those references should be cited.

Response: We are not stating that there have not been size-segregated studies of dicarboxylic acids, we are saying there are not that many size-segregated studies of dicarboxylic acids that are long term (> 6 months) with high sampling frequency (at least weekly), and with temporal resolution around ≤ 48-hours. The references that were shared here do not meet most if not all of those criteria, therefore, we do not think it is necessary to add in those references. The manuscript already has 215 references.

4. Method section. There is no description on the sample storage from sampling to chemical analysis. Did you store the samples at room temperature or at -20°C in a freezer? After the water extraction of the sample, how long did you store the water extracts at room temperature? Storage of filter samples and water extracts at room temperature may be subject to biodegradation of organic acids. In Manila, high humidity conditions may provide more moistures to the aerosols. Please provide the information of storage of samples and extracts.

Response: Once samples were collected they were sealed in a Petrislide and wrapped with Parafilm before being stored in a freezer at – 20 °C. After water extractions the samples were immediately analyzed using ion chromatography, limiting the amount of biodegradation. Storage and extraction methods have been described in great detail for these samples in Stahl et al. (2020), however, a brief description of these methods was added for clarification:

Lines 166-172: "Details of the sample sets are shown in Table S2 can be found in more detail in Stahl et al. (2020), but a brief summary of the storage and extraction methods will be described here. Substrates were stored in a freezer at -20 °C after samples were collected from the MOUDI until extractions could be carried out. The stored substrates were then extracted by sonication in Milli-Q water (18.2 MΩ-cm) for 30 minutes. After sonication, solutions were immediately analyzed to prevent degradation while the remaining extracts were stored in a refrigerator for additional analyses."

5. Results section. Some paragraphs are too long and redundant (e.g., 31 lines in the paragraph starting at line 253). By reorganizing the paragraphs and rephrasing the sentences, authors could improve them to become more readable.

Response: The paragraph was broken into 3 separate paragraphs (MSA, saturated organic acids, and unsaturated organic acids).

6. Lines 340-344. Authors mentioned lower concentrations of oxalate and MSA than expected for a megacity Manila. This point should be critically verified including the discussion of methods used; potential biodegradation of organic acids and MSA during sample storage and analytical protocol.

Response: As stated previously a section has been added to address the methods used. Biodegradation of the organic acids and MSA is very possible in the environment, however, measures were taken to ensure samples were properly stored to minimize the potential of biodegradation of the samples. Sample filters were stored in a freezer after they were collected and were extracted once they were going to be analyzed. Therefore, we do not believe this comments warrants additional revision.

7. Discussion. What is the reason to start the discussion with phthalate that is not the major organic acid? Why do you discuss oxalate at the end that is the most abundant organic acid? I wondered if you could improve the discussion section by reorganizing the order of the compounds.
Response: The reasoning was to start with the largest chain organic (phthalate) and end with the smallest (oxalate), followed by MSA. This is because the large chain organics will decompose to the smaller chain organics and we thought it was necessary to discuss the larger species first to properly relay the narrative of where the smaller chain organic acids come from. For example, it helps in the discussion of oxalate to have first covered the larger acids that can break down to form oxalate. We revised this line:

Lines 475-477: "Subsequent sections discuss each organic acid and MSA in more detail, beginning with larger acids since knowledge of their behavior is important to better understand the smaller acids."

8. Lines 514-515. How do you explain the crustal sources (35.9%) for adipate? In other places in the text, authors discuss the combustion sources of adipic acid (lines 261-262).
Response: The crustal sources for adipate are likely due to adsorption onto larger dust particles. We are not saying that adipate comes from crustal sources, rather we are saying it is added on through aging. There are no plausible explanations of direct emissions for supermicrometer adipate to our knowledge, so the only explanation is that adipate (and other organic acids for that matter) adsorbs onto larger particles. Work by Sulliavan and Prather (2007) further implies that dicarboxylic acids have an affinity towards adsorption onto dust particles over sea salt. This reinforces our stance that adipate is adsorbing onto these larger dust particles, which is why adipate is associated strongly with crustal sources.

9. Lines 621-622 and 704-706. Again how do you explain the "surprisingly low concentrations of oxalate" ? This reviewer is skeptical about data quality; i.e., a potential loss of organic acids and MSA due to biodegradation during the sample storage and analytical procedure used in this study.
Response: As stated multiple time above, once the samples were removed from the MOUDI they were sealed and stored in a freezer until they were ready to be analyzed. Then they were extracted and immediately analyzed with little to no storage time.

10. Lines 695-698. Is it possible to discuss the fraction of organic acids and MSA on a carbon basis? Did you measure TOC or WSOC?
Response: Unfortunately, TOC and WSOC were not measured so we are not able to discuss the fraction of organic acids and MSA on a carbon basis.

**References**
Stahl, C., Cruz, M. T., Banaga, P. A., Betito, G., Braun, R. A., Aghdam, M. A., Cambaliza, M. O., Lorenzo, G. R., MacDonald, A. B., Pabroa, P. C., Yee, J. R., Simpas, J. B., and Sorooshian, A.: An annual time series of weekly size-resolved aerosol properties in the megacity of Metro Manila, Philippines, Sci Data, 7, 128, 10.1038/s41597-020-0466-y, 2020.

---

## Author Response (AR2)

| 1
2
3
4
5                                 | Response: We thank the editor for thoughtful suggestions and constructive criticism that have helped us improve our manuscript. Below we provide responses to the remaining concerns and suggestions in blue font. All changes to the manuscript can be identified in the version submitted using Track Changes.                                                                                                                                                                                                                                                                                                                                                                                                                                                                                                  |
|-------------------------------------------------------|-------------------------------------------------------------------------------------------------------------------------------------------------------------------------------------------------------------------------------------------------------------------------------------------------------------------------------------------------------------------------------------------------------------------------------------------------------------------------------------------------------------------------------------------------------------------------------------------------------------------------------------------------------------------------------------------------------------------------------------------------------------------------------------------------------------------|
| 6                                                     | Editor Response:                                                                                                                                                                                                                                                                                                                                                                                                                                                                                                                                                                                                                                                                                                                                                                                                  |
| 7

16 | <ol> <li>I agree with Referee 2 that further details are required regarding the analytical procedure and
that further explanation for the low concentrations needs to be provide. The addition of the text
regarding sample storage is good, however a few more details are required. How long were the
samples typically frozen before analysis? Additionally sonication can create reactive oxygen
species (e.g., Miljevic et al., 2014) and degrade organic compounds. Was this investigated? Does
this have any influence on the low concentrations (relative to other polluted areas) observed?</li> <li>Additionally, I agree with Referee 2 that there needs to be more discussion about possible
reasons for the low concentrations, particularly for oxalate and MSA.</li> </ol> |
| 17
18
19
20
21
22                      | Response: Samples, on average, were stored for 2 weeks. Additional text has been added to say this. The effects of sonication were investigated for the six species in question and it was found that there was no significant change in the concentration. The text has been updated to say so. In summary, sonication is not a likely reason why the concentrations are so low compared to other regions. An additional paragraph was added at the end of the MSA section in the discussion to discuss possible reasons for the low concentrations.                                                                                                                                                                                                                                                             |
| 23                                                    | Line 170: " carried out, which on average was approximately 2 weeks."                                                                                                                                                                                                                                                                                                                                                                                                                                                                                                                                                                                                                                                                                                                                             |
| 24
25
26
27                                  | Line 173-176: "There have been studies that discuss the effects of sonication oxidation degrading organic species (i.e., Miljevic et al. 2014). It was determined through experimental tests that no significant degradation occurred during the sonication process for the species being analyzed in this study."                                                                                                                                                                                                                                                                                                                                                                                                                                                                                                |
| 28

36    | Lines 726-734: "Both MSA and oxalate had significantly lower concentrations than other regions, and there are a few possible explanations for this. First, it is worth noting that degradation of these species is unlikely due to storage or sonication as careful procedures were followed as noted in Sect. 2.2. The Philippines has relatively high temperatures, humidity, and solar radiation year-round, providing optimal conditions for processing and degradation to occur, yielding low concentrations for MSA and oxalate. Furthermore, there are mechanisms by which species such as oxalate can be degraded via complexation effects with metal cations (Paris and Desboeufs, 2013; Siffert and Sulzberger, 1991; Sorooshian et al., 2013; Zuo, 1995), which are abundant in the study region."     |
| 37                                                    |                                                                                                                                                                                                                                                                                                                                                                                                                                                                                                                                                                                                                                                                                                                                                                                                                   |
| 38                                                    |                                                                                                                                                                                                                                                                                                                                                                                                                                                                                                                                                                                                                                                                                                                                                                                                                   |

39 2) I agree with Referee 1's that the fact that the MSA PMF results had 0% from sea salt needs to

- 40 be discussed in more detail. In particular, the idea of fresh vs processed sea salt should be
- 41 explored in the manuscript, not just in the response to referee comments.
- 42
- Response: Additional text has been added to address the fact that MSA is not associated with seasalt in the PMF.
- 45 Lines 716-718: "Due to the proximity of the sampling site to the ocean it is possible that the local
- 46 sea salt was relatively fresh with short transport time, which could potentially explain the lack of
- 47 an association with MSA as it requires time to be produced from its marine precursor DMS."
- 48 Lines 721-723: "Consequently, concentrations of MSA from these other non-marine sources
- 49 could be much higher causing the PMF model to associate MSA with non-sea salt related
- 50 sources."
- 51
- 52 3) I agree with Referee 2's second comment that the citation of Kondo et al (2011) is not
- 53 warranted in this location (line 52-53 of track changes version). The sentence is specifically
- 54 discussing organic acids, not organic aerosol more generally. While Kondo et al (2011) describe
- organic aerosol, the manuscript does not include organic acids. I believe this is the case for
- 56 several of the references provided for this opening sentence.
- 57
- 58 Response: The references (Ding et al., 2013, Duarte et al., 2017, Kondo et al., 2011, Skyllakou et
- al., 2017, Sun et al., 2012, and Youn et al., 2013) that do not discuss organic acids in the first
- 60 sentence have been removed. An additional reference has been added that fit more appropriately
- 61 (Kawamura et al., 2005). The first sentence now reads:
- 62 Lines 50-52: "Organic acids are ubiquitous components of ambient particulate matter and can
- 63 contribute appreciably to total mass concentrations in diverse regions ranging from the Arctic to
- 64 deserts (e.g., Barbaro et al., 2017; Gao et al., 2003; Kawamura et al., 2005)."
- 65
- 4) I agree with Referee 1's comment 8 that the long list of measurements in other locations is notneeded. It is difficult to read, particularly when measurements from various environments are
- 68 compared (i.e. Arctic) where one would expect extremely different concentrations. It is hard for
- 69 the reader to judge if concentrations are truly similar or not. I recommend only including the
- 70 references to similar environments in the main text and moving additional locations to the SI. It
- 71 would also be useful to report the concentrations in the same units (ng/m3 or ug/m3) for a given
- 72 acid as this would aid in quick comparison.
- 73
- 74 Response: The list of measurements for each species have been removed. Oxalate and MSA
- 75 include a list of measurements but only from Asia as those will be the most relatable. Two
- 76 studies were swapped for oxalate to be more relatable to the region. Any measurements in  $\mu g$
- 77 were converted to ng to keep the units consistent.

78 Lines 651-652: "... 195 – 669 ng m-3 in Beijing, China (Du et al., 2014); and 149 - 735 ng m-3 in

79 Thumba, India (Hegde et al., 2016)."

80

- 5) There are several instances where the strength of the wording needs to be adjusted. For
- 82 instance, line 666-667 (track changes) "... Translation period indicate that photooxidation..." I
- think "suggests" is a better word than "indicates" since this is based largely on correlation not on
- 84 a complete oxidative analysis. The use of "indicate" is also perhaps too strong on line 666.
- 85 Similarly in the abstract (42-43), it is more appropriate to say that the results are consistent with
- 86 gas-to-particle conversion rather than "... gas-to-particle conversion largely explained ..." That
- 87 would require measurements of the gases. There may be other instances as well.
- 88
- 89 Response: This has been addressed:
- 90 Lines 42-43: "... formation via gas-to-particle conversion largely explained..." has been
- 91 changed to "... formation via gas-to-particle conversion is consistent with submicrometer peaks
- 92 for the organic acids and MSA..."
- 93 Line 48: "Indicates" has been switched to "suggests".
- 94 "... suggests photo-oxidation is an important formation pathway."
- 95 Line 434: "Indicate" has not been changed as it is concluded from the results.
- 96 "The supermicrometer results indicate MSA was correlated only with Na (r = 0.32),
  97 due..."
- 98 Line 619: "Indicates" has been changed to "suggests".
- 99 "... most other species, suggests that it had less diverse sources and was derived from100 combustion..."
- 101 Line 670: "Indicate" has been changed to "suggest".

102 "Additionally, high concentrations of oxalate in the Transitional period suggest that..."

- 103 Line 680: "Indicate" has been switched to "suggest".
- 104 "The PMF results from the present study suggest that..."

- 1 Sources and characteristics of size-resolved particulate organic acids and methanesulfonate in a
- 2 coastal megacity: Manila, Philippines
- 3 Connor Stahl1, Melliza Templonuevo Cruz2,3, Paola Angela Bañaga2,4, Grace Betito2,4, Rachel A.
- 4 Braun1, Mojtaba Azadi Aghdam1, Maria Obiminda Cambaliza2,4, Genevieve Rose Lorenzo2,5,
- 5 Alexander B. MacDonald1, Miguel Ricardo A. Hilario2, Preciosa Corazon Pabroa6, John Robin
- 6 Yee6, James Bernard Simpas2,4, Armin Sorooshian1,5

[revised manuscript text omitted]

| Size/Species |           | All (n         | = 54)     | SWM18          | (n = 11)   | Transitiona     | l (n = 3) | NEM (r         | n = 27)     | SWM19 (n = 13) |           |  |
|--------------|-----------|----------------|-----------|----------------|------------|-----------------|-----------|----------------|-------------|----------------|-----------|--|
| 512          | e/species | Range          | Mean (SD) | Range          | Mean (SD)  | Range           | Mean (SD) | Range          | Mean (SD)   | Range          | Mean (SD) |  |
| m            | Phthalate | 0 - 67.02      | 9 (14)    | 1.97 - 67.02   | 17 (25)    | 17.36 - 45.30   | 27 (16)   | 0 - 14.72      | 4.8 (4.4)   | 0-25.03        | 5.7 (7.4) |  |
| щ 8          | Adipate   | 0 - 43.83      | 7.6 (9.4) | 0 - 20.18      | 9.1 (8.8)  | 0.24 - 19.56    | 8 (10)    | 0 - 13.00      | 4.2 (3.8)   | 0 - 43.83      | 13 (15)   |  |
| 5 - 1        | Succinate | 0 - 116.28     | 10 (22)   | 0 - 116.28     | 22 (43)    | 0 - 14.31       | 7.6 (7.2) | 0 - 62.83      | 7 (14)      | 0 - 20.14      | 4.7 (7.4) |  |
| .056         | Maleate   | 0 - 119.19     | 10 (20)   | 2.56 - 58.39   | 19 (15)    | 0.19 - 8.45     | 3.8 (4.2) | 0 - 14.42      | 1.7 (3.7)   | 2.30 - 119.19  | 19 (34)   |  |
| 1: 0         | Oxalate   | 37.67 - 472.82 | 149 (94)  | 49.83 - 472.82 | 178 (139)  | 179.42 - 365.10 | 252 (99)  | 51.62 - 421.82 | 144 (76)    | 37.67 - 214.62 | 110 (62)  |  |
| AI           | MSA       | 0.10 - 23.33   | 5.4 (5.2) | 2.77 - 23.33   | 10.0 (6.6) | 0.16 - 16.14    | 5.6 (9.2) | 0.10 - 7.45    | 3.1 (2.0)   | 0.84 - 17.52   | 6.3 (5.4) |  |
|              | Phthalate | 0 - 64.53      | 6 (13)    | 0.51 - 64.53   | 15 (24)    | 9.14 - 39.62    | 20 (17)   | 0 - 9.38       | 1.6 (2.5)   | 0 - 8.51       | 2.7 (3.1) |  |
| E            | Adipate   | 0 - 31.57      | 4.3 (5.8) | 0 - 15.94      | 6.1 (6.3)  | 0 - 10.99       | 5.5 (5.5) | 0 - 10.64      | 2.5 (3.2)   | 0 - 31.57      | 6.1 (8.8) |  |
| - 1-
-    | Succinate | 0 - 108.47     | 7 (20)    | 0 - 108.47     | 19 (39)    | 0 - 13.54       | 7.3 (6.8) | 0 - 52.42      | 4 (10)      | 0 - 15.68      | 4.3 (6.6) |  |
| 56 -         | Maleate   | 0 - 108.65     | 9 (18)    | 2.56 - 57.73   | 18 (15)    | 0.19 - 8.45     | 3.8 (4.2) | 0 - 14.42      | 1.6 (3.6)   | 2.30 - 108.65  | 18 (31)   |  |
| 0.0          | Oxalate   | 16.21 - 318.49 | 93 (62)   | 29.96 - 256.72 | 108 (75)   | 96.84 - 250.78  | 166 (78)  | 26.11 - 318.49 | 91 (58)     | 16.21 - 151.79 | 70 (40)   |  |
|              | MSA       | 0 - 21.32      | 5.0 (4.9) | 2.41 - 21.32   | 9.3 (6.2)  | 0.08 - 15.58    | 5.3 (8.9) | 0 - 7.45       | 2.9 (2.1)   | 0.84 - 16.22   | 5.7 (5.1) |  |
|              | Phthalate | 0 - 16.52      | 3.1 (3.3) | 0 - 4.07       | 2.0 (1.7)  | 5.43 - 9.03     | 6.7 (2.0) | 0 - 9.42       | 3.2 (2.6)   | 0 - 16.52      | 3.0 (5.0) |  |
| _            | Adipate   | 0 - 26.00      | 3.3 (4.9) | 0 - 7.87       | 3.0 (3.2)  | 0 - 8.56        | 2.9 (4.9) | 0 - 8.07       | 1.7 (2.2)   | 0 - 26.00      | 7.1 (8.0) |  |
| un           | Succinate | 0 - 21.18      | 2.2 (4.5) | 0 - 16.02      | 3.1 (4.9)  | 0 - 0.77        | 0.3 (0.4) | 0 - 21.18      | 2.9 (5.4)   | 0 - 5.33       | 0.4 (1.5) |  |
| -18          | Maleate   | 0 - 10.54      | 0.4 (1.5) | 0 - 2.30       | 0.3 (0.7)  | 0               | 0         | 0 - 0.45       | 0.02 (0.09) | 0 - 10.54      | 1.2 (2.9) |  |
| _            | Oxalate   | 6.27 - 216.10  | 55 (39)   | 19.87 - 216.10 | 70 (67)    | 62.90 - 114.32  | 87 (26)   | 18.51 - 104.88 | 53 (23)     | 6.27 - 103.58  | 41 (29)   |  |
|              | MSA       | 0 - 2.00       | 0.4 (0.5) | 0 - 2.00       | 0.8 (0.6)  | 0 - 0.56        | 0.2 (0.3) | 0 - 1.58       | 0.2 (0.4)   | 0 - 1.93       | 0.6 (0.6) |  |

**Table 1:** Seasonal concentrations (ng m-3) of organic acids and MSA for all  $(0.056 - 18 \,\mu\text{m})$ , submicrometer  $(0.056 - 1 \,\mu\text{m})$ , and supermicrometer  $(1 - 18 \,\mu\text{m})$  sizes measured in Metro Manila from July 2018 to October 2019. n = number of sets.

| 1552 | Table 2: Contributions of the five positive matrix factorization (PMF) source factors to each |
|------|-----------------------------------------------------------------------------------------------|
| 1553 | individual organic acid and MSA.                                                              |

|           | Combustion | Biomass
Burning | Crustal | Sea
Salt | Waste
Processing |
|-----------|------------|--------------------|---------|-------------|---------------------|
| Phthalate | 27.4 %     | 49.5 %             | 13.3 %  | 9.9 %       | 0 %                 |
| Adipate   | 32.9 %     | 26.4 %             | 35.9 %  | 4.7 %       | 0 %                 |
| Succinate | 0 %        | 90.3 %             | 9.7 %   | 0 %         | 0 %                 |
| Maleate   | 69.7 %     | 0 %                | 0.2 %   | 0 %         | 30.1 %              |
| Oxalate   | 32.9 %     | 25.4 %             | 31.2 %  | 0 %         | 10.5 %              |
| MSA       | 57.4 %     | 41.2 %             | 0.1 %   | 0 %         | 1.4 %               |

- **Table 3:** Pearson's correlation matrices (r values) of water-soluble species for submicrometer
- $(0.056 1.0 \ \mu\text{m})$  and supermicrometer  $(1.0 18 \ \mu\text{m})$  sizes. Blank boxes indicate p-values
- 1558 exceeding 0.05 and thus deemed to be statistically insignificant. Ad adipate, Su succinate,
- 1559 Ma maleate, Ox oxalate, Ph phthalate. A similar correlation matrix for the full size range
- $(0.056 18 \,\mu\text{m})$  is in Table S6.

| Al                                                                                                              | 1.00                                                                         |                                                              |                      |                                                      |                                      |                      |              |              |      |                                                      |                      |                      |            |      |                      |      |      |      |      |
|-----------------------------------------------------------------------------------------------------------------|------------------------------------------------------------------------------|--------------------------------------------------------------|----------------------|------------------------------------------------------|--------------------------------------|----------------------|--------------|--------------|------|------------------------------------------------------|----------------------|----------------------|------------|------|----------------------|------|------|------|------|
| Ti                                                                                                              |                                                                              | 1.00                                                         |                      |                                                      |                                      |                      |              |              |      |                                                      |                      |                      |            |      |                      |      |      |      |      |
| К                                                                                                               | 0.91                                                                         |                                                              | 1.00                 |                                                      |                                      |                      |              |              |      |                                                      |                      |                      |            |      |                      |      |      |      |      |
| Rb                                                                                                              | 0.44                                                                         |                                                              | 0.48                 | 1.00                                                 |                                      |                      |              |              |      |                                                      |                      |                      |            |      |                      |      |      |      |      |
| V                                                                                                               |                                                                              | 0.28                                                         |                      | 0.36                                                 | 1.00                                 |                      |              |              |      |                                                      |                      |                      |            |      |                      |      |      |      |      |
| Ni                                                                                                              |                                                                              | 0.47                                                         |                      | 0.40                                                 | 0.89                                 | 1.00                 |              |              |      |                                                      |                      |                      |            |      |                      |      |      |      |      |
| As                                                                                                              |                                                                              |                                                              |                      |                                                      |                                      |                      | 1.00         |              |      |                                                      |                      |                      |            |      |                      |      |      |      |      |
| Cd                                                                                                              |                                                                              |                                                              |                      |                                                      | 0.64                                 | 0.68                 |              | 1.00         |      |                                                      |                      |                      |            |      |                      |      |      |      |      |
| Pb                                                                                                              | 0.41                                                                         |                                                              | 0.32                 | 0.27                                                 | 0.28                                 | 0.40                 |              | 0.42         | 1.00 |                                                      |                      |                      |            |      |                      |      |      |      |      |
| Na                                                                                                              |                                                                              |                                                              |                      |                                                      |                                      |                      |              |              |      | 1.00                                                 |                      |                      |            |      |                      |      |      |      |      |
| Cl                                                                                                              | 0.90                                                                         |                                                              | 0.99                 | 0.39                                                 |                                      |                      |              |              | 0.30 |                                                      | 1.00                 |                      |            |      |                      |      |      |      |      |
| NO3                                                                                                             | 0.76                                                                         |                                                              | 0.82                 | 0.28                                                 |                                      |                      |              |              |      |                                                      | 0.84                 | 1.00                 | ]          |      |                      |      |      |      |      |
| SO4                                                                                                             |                                                                              |                                                              |                      | 0.42                                                 | 0.48                                 | 0.40                 |              |              |      |                                                      |                      |                      | 1.00       |      |                      |      |      |      |      |
| MSA                                                                                                             |                                                                              |                                                              |                      | 0.39                                                 |                                      |                      |              |              |      |                                                      |                      |                      | 0.60       | 1.00 | ]                    |      |      |      |      |
| Ad                                                                                                              |                                                                              |                                                              |                      |                                                      |                                      |                      |              |              |      |                                                      |                      |                      |            |      | 1.00                 |      |      |      |      |
| Su                                                                                                              |                                                                              | 0.31                                                         |                      | 0.67                                                 |                                      |                      |              |              |      |                                                      |                      |                      | 0.45       | 0.67 | 0.33                 | 1.00 | ]    |      |      |
| Ma                                                                                                              |                                                                              |                                                              |                      |                                                      |                                      |                      |              |              |      |                                                      |                      |                      |            |      | 0.32                 |      | 1.00 |      |      |
| Ox                                                                                                              |                                                                              | 0.35                                                         |                      | 0.70                                                 | 0.47                                 | 0.53                 |              |              |      |                                                      |                      |                      | 0.72       | 0.47 |                      | 0.69 |      | 1.00 |      |
| Ph                                                                                                              |                                                                              | 0.37                                                         |                      | 0.53                                                 |                                      |                      |              |              |      |                                                      |                      |                      | 0.39       | 0.67 | 0.45                 | 0.82 |      | 0.57 | 1.00 |
|                                                                                                                 | Al                                                                           | Ti                                                           | K                    | Rb                                                   | v                                    | Ni                   | As           | Cd           | Pb   | Na                                                   | Cl                   | NO3                  | SO4 | MSA  | Ad                   | Su   | Ma   | Ox   | Ph   |
| \1.um                                                                                                    | 1                                                                            |                                                              |                      |                                                      |                                      |                      |              |              |      |                                                      |                      |                      |            |      |                      |      |      |      |      |
| × 1 µm                                                                                                          |                                                                              |                                                              |                      |                                                      |                                      |                      |              |              |      |                                                      |                      |                      |            |      |                      |      |      |      |      |
| Δ1                                                                                                              | 1.00                                                                         | 1                                                            |                      |                                                      |                                      |                      |              |              |      |                                                      |                      |                      |            |      |                      |      |      |      |      |
| Al
Ti                                                                                                        | 1.00                                                                         | 1.00                                                         | ]                    |                                                      |                                      |                      |              |              |      |                                                      |                      |                      |            |      |                      |      |      |      |      |
| Al
Ti
K                                                                                                   | 1.00
0.56                                                                 | 1.00                                                         | 1.00                 |                                                      |                                      |                      |              |              |      |                                                      |                      |                      |            |      |                      |      |      |      |      |
| Al
Ti
K
Rb                                                                                             | 1.00
0.56                                                                 | 1.00                                                         | 1.00                 | 1.00                                                 | ]                                    |                      |              |              |      |                                                      |                      |                      |            |      |                      |      |      |      |      |
| Al
Ti
K
Rb
V                                                                                        | 1.00
0.56
0.62                                                         | 1.00                                                         | 1.00
0.48         | 1.00                                                 | 1.00                                 |                      |              |              |      |                                                      |                      |                      |            |      |                      |      |      |      |      |
| Al
Ti
K
Rb
V                                                                                        | 1.00
0.56
0.62                                                         | 1.00
0.40                                                 | 1.00
0.48         | 1.00
0.31                                         | 1.00                                 | 1.00                 |              |              |      |                                                      |                      |                      |            |      |                      |      |      |      |      |
| Al
Ti
K
Rb
V
Ni
As                                                                            | 1.00
0.56
0.62                                                         | 1.00
0.40
0.30
0.37                                 | 1.00 0.48            | 1.00
0.31                                         | 1.00                                 | 1.00                 | 1.00         | 1            |      |                                                      |                      |                      |            |      |                      |      |      |      |      |
| Al
Ti
K
Rb
V
Ni
As
Cd                                                                      | 1.00
0.56
0.62                                                         | 1.00
0.40
0.30
0.37                                 | 1.00 0.48            | 1.00                                                 | 1.00
0.33
0.66                 | 1.00                 | 1.00
0.34 | 1.00         | l    |                                                      |                      |                      |            |      |                      |      |      |      |      |
| Al
Ti
K
Rb
V
Ni
As
Cd
Pb                                                                | 1.00
0.56
0.62                                                         | 1.00
0.40
0.30
0.37
0.45                         | 1.00 0.48            | 1.00
0.31
0.36                                 | 1.00
0.33
0.66
0.51         | 1.00
0.41
0.45 | 1.00
0.34 | 1.00
0.65 | 1.00 | 1                                                    |                      |                      |            |      |                      |      |      |      |      |
| Al
Ti
K
Rb
V
Ni
As
Cd
Pb
Na                                                          | 1.00
0.56
0.62
0.43
0.43                                         | 1.00
0.40
0.30
0.37
0.45
0.42                 | 1.00 0.48            | 1.00
0.31
0.36                                 | 1.00
0.33
0.66
0.51         | 1.00
0.41
0.45 | 1.00
0.34 | 1.00
0.65 | 1.00 | 1.00                                                 | I                    |                      |            |      |                      |      |      |      |      |
| Al
Ti
K
Rb
V
Ni
As
Cd
Pb
Na                                                          | 1.00
0.56
0.62
0.43
0.43
0.49
0.45                         | 1.00
0.40
0.30
0.37
0.45
0.42
0.48         | 1.00 0.48            | 1.00
0.31
0.36                                 | 1.00
0.33
0.66
0.51         | 1.00
0.41
0.45 | 1.00
0.34 | 1.00
0.65 | 1.00 | 1.00
0.90                                         | 1.00                 | Ι                    |            |      |                      |      |      |      |      |
| Al
Ti
K
Rb
V
Ni
As
Cd
Pb
Na
Cl
NO3                                             | 1.00
0.56
0.62
0.43
0.43
0.49
0.45
0.38                 | 1.00
0.40
0.30
0.37
0.45
0.42
0.48         | 1.00 0.48            | 1.00
0.31
0.36                                 | 1.00
0.33
0.66
0.51         | 1.00
0.41
0.45 | 1.00
0.34 | 1.00
0.65 | 1.00 | 1.00
0.90
0.64                                 | 1.00
0.30         | 1.00                 | 1          |      |                      |      |      |      |      |
| AI
Ti
K
Rb
V
Ni
As
Cd
Pb
Na
Cl
NO3
SO4                                      | 1.00
0.56
0.62
0.43
0.43
0.49
0.45
0.38
0.39         | 1.00
0.40
0.30
0.37
0.45
0.42
0.48         | 1.00 0.48 0.81       | 1.00
0.31
0.36
0.32
0.64                 | 1.00
0.33
0.66
0.51         | 1.00
0.41
0.45 | 1.00         | 1.00 0.65    | 1.00 | 1.00
0.90
0.64
0.37                         | 1.00
0.30
0.29 | 1.00
0.36         | 1.00       | 1    |                      |      |      |      |      |
| AI
Ti
K
Rb
V
Ni
As
Cd
Pb
Na
Cl
NO3
SO4
MSA                               | 1.00
0.56
0.62
0.43
0.43
0.49
0.45
0.38
0.39         | 1.00
0.40
0.30
0.37
0.45
0.42
0.48         | 1.00
0.48         | 1.00
0.31
0.36
0.32
0.64                 | 1.00
0.33
0.66
0.51         | 1.00
0.41
0.45 | 1.00 0.34    | 1.00         | 1.00 | 1.00
0.90
0.64
0.37
0.32                 | 1.00
0.30
0.29 | 1.00
0.36         | 1.00       | 1.00 | 1                    |      |      |      |      |
| AI
Ti
K
Rb
V
Ni
As
Cd
Pb
Na
Cl
NO3
SO4
MSA
Ad                         | 1.00
0.56
0.62
0.43
0.49
0.45
0.38
0.39                 | 1.00
0.40
0.30
0.37
0.45
0.42
0.48         | 1.00
0.48         | 1.00
0.31
0.36
0.32
0.64                 | 1.00
0.33
0.66
0.51
0.41 | 1.00
0.41
0.45 | 1.00 0.34    | 1.00 0.65    | 1.00 | 1.00
0.90
0.64
0.37
0.32                 | 1.00
0.30
0.29 | 1.00
0.36         | 1.00       | 1.00 | 1.00                 | 1    |      |      |      |
| Al
Ti
K
Rb
V
Ni
As
Cd
Pb
Na
Cl
NO3
SO4
MSA
Ad
Su                   | 1.00
0.56
0.62
0.43
0.49
0.45
0.38
0.39                 | 1.00
0.40
0.30
0.37
0.45
0.42
0.48         | 1.00
0.48
0.81 | 1.00
0.31
0.36
0.32
0.64                 | 1.00
0.33
0.66
0.51         | 1.00
0.41
0.45 | 1.00 0.34    | 1.00 0.65    | 1.00 | 1.00
0.90
0.64
0.37
0.32                 | 1.00
0.30
0.29 | 1.00 0.36            | 1.00       | 1.00 | 1.00                 | 1.00 | 1    |      |      |
| AI
Ti
K
Rb
V
Ni
As
Cd
Pb
Na
Cl
NO3
SO4
MSA
Ad
Su
Ma             | 1.00
0.56
0.62
0.43
0.49
0.45
0.38
0.39
0.39         | 1.00
0.40
0.30
0.45
0.45
0.42
0.48         | 1.00
0.48
0.81 | 1.00
0.31
0.36
0.32
0.64                 | 1.00
0.33
0.66
0.51
0.41 | 1.00
0.41
0.45 | 1.00 0.34    | 1.00 0.65    | 1.00 | 1.00
0.90
0.64
0.37
0.32
0.30         | 1.00
0.30
0.29 | 1.00 0.36            | 1.00       | 1.00 | 1.00                 | 1.00 | 1.00 |      |      |
| AI
Ti
K
Rb
V
Ni
As
Cd
Pb
Na
Cl
NO3
SO4
MSA
Ad
Su
Ma
Ox       | 1.00
0.56
0.62
0.43
0.49
0.45
0.38
0.39
0.39
0.39 | 1.00
0.40
0.30
0.37
0.45
0.42
0.48         | 1.00
0.48         | 1.00
0.31
0.36
0.32
0.64
0.28
0.28 | 1.00
0.33
0.66
0.51
0.41 | 1.00
0.41
0.45 | 1.00 0.34    | 1.00 0.65    |      | 1.00
0.90
0.64
0.37
0.32
0.30
0.45 | 1.00
0.30
0.29 | 1.00
0.36         | 1.00       | 1.00 | 1.00                 | 1.00 | 1.00 | 1.00 |      |
| Al
Ti
K
Rb
V
Ni
As
Cd
Pb
Na
Cl
NO3
SO4
MSA
Ad
Su
Ma
Ox
Ph | 1.00
0.56
0.62
0.43
0.49
0.45
0.38
0.39
0.39
0.39 | 1.00
0.40
0.30
0.45
0.45
0.42
0.48
0.48 | 0.81                 | 1.00
0.31
0.36
0.32
0.64
0.28
0.48 | 1.00
0.33
0.66
0.51         | 1.00
0.41
0.45 |              | 1.00 0.65    |      | 1.00
0.90
0.64
0.37
0.32
0.30
0.45 | 1.00
0.30
0.29 | 1.00
0.36
0.59 | 1.00       | 1.00 | 1.00
0.57
0.30 | 1.00 | 1.00 | 1.00 | 1.00 |